# Hippo signaling determines the number of venous pole cells that originate from the anterior lateral plate mesoderm in zebrafish

Hajime Fukui[1,2,3,4,5,6], Takahiro Miyazaki[1], Renee Wei-Yan Chow[3,4,6,5], Hiroyuki Ishikawa[1], Hiroyuki Nakajima[1], Julien Vermot[3,4,6,5], Naoki Mochizuki[1,7]*

[1]Department of Cell Biology, National Cerebral and Cardiovascular Center Research Institute, Suita, Japan; [2]University of Strasbourg Institute for Advanced Study (USIAS), Strasbourg, France; [3]Institut de Génétique et de Biologie Moléculaire et Cellulaire, Illkirch, France; [4]Centre National de la Recherche Scientifique, Illkirch, France; [5]Institut National de la Santé et de la Recherche Médicale, Illkirch, France; [6]Université de Strasbourg, Illkirch, France; [7]AMED-Core Research for Evolutional Science and Technology (AMED-CREST), Japan Agency for Medical Research and Development (AMED), Tokyo, Japan

**Abstract** The differentiation of the lateral plate mesoderm cells into heart field cells constitutes a critical step in the development of cardiac tissue and the genesis of functional cardiomyocytes. Hippo signaling controls cardiomyocyte proliferation, but the role of Hippo signaling during early cardiogenesis remains unclear. Here, we show that Hippo signaling regulates atrial cell number by specifying the developmental potential of cells within the anterior lateral plate mesoderm (ALPM), which are incorporated into the venous pole of the heart tube and ultimately into the atrium of the heart. We demonstrate that Hippo signaling acts through *large tumor suppressor kinase 1/2* to modulate BMP signaling and the expression of *hand2*, a key transcription factor that is involved in the differentiation of atrial cardiomyocytes. Collectively, these results demonstrate that Hippo signaling defines venous pole cardiomyocyte number by modulating both the number and the identity of the ALPM cells that will populate the atrium of the heart.
DOI: https://doi.org/10.7554/eLife.29106.001

*For correspondence: mochizuki@ncvc.go.jp

## Introduction

The human heart typically has about 2 billion cardiomyocytes (CMs) (*Adler and Costabel, 1975*; *Laflamme and Murry, 2011*), which together form the muscle layer of the heart responsible for contraction. The determination of the final number of CMs in the different parts of the heart involves highly coordinated processes of cell fate specification and proliferation during development. Understanding the relative contributions of these processes during the different stages of cardiac morphogenesis, as well as the mechanisms behind them, is one of the long-standing goals of the cardiac development field.

Mammalian heart morphogenesis is best studied in the mouse. Early in mouse development, a bilateral group of cells in the splanchnic mesoderm specifies into cardiac precursor cells (CPCs) (*Saga et al., 1999*) and forms the first heart field (FHF). Cells of the FHF then extend toward the midline to form a crescent-shaped epithelium, known as the cardiac crescent. Through a series of morphogenetic steps, the cardiac crescent gives rise to a structure called the heart tube (*Kelly et al., 2014*; *Vincent and Buckingham, 2010*). CPCs from the secondary heart field (SHF),

which are derived from pharyngeal mesoderm, are added to the arterial and venous poles of the heart tube (*Cai et al., 2003*; *Kelly et al., 2001*; *Waldo et al., 2001*). The FHF is believed to give rise mostly to the left ventricle and parts of the atria, whereas the SHF is believed to give rise mostly to the right ventricle, the outflow tract (OFT) and most of the atria (*Cai et al., 2003*; *Galli et al., 2008*; *Waldo et al., 2001*; *Zaffran et al., 2004*).

The zebrafish has a simpler heart than that of mouse and humans, containing only two chambers. Nevertheless, the successive phases of CM differentiation during development, as well as their associated genetic pathways, are well conserved between zebrafish and other vertebrates (*Staudt and Stainier, 2012*). Because of the optical clarity and external development of zebrafish embryos, as well as the amenability of zebrafish to genetic manipulation, the zebrafish is an excellent model for the study of cardiac development. In the early stages of zebrafish development, a bilateral group of cells in the anterior lateral plate mesoderm (ALPM) specifies into CPCs, and the region where they reside is termed the heart field (HF) (*Fishman and Chien, 1997*). In zebrafish, both the FHF and the SHF derive from the ALPM (*Mosimann et al., 2015*). As in the mouse, the zebrafish FHF forms the initial heart tube while the SHF elongates the heart tube by adding to its arterial and venous poles. LIM domain transcription factor Islet1 (Isl1) marks a subset of SHF cells (*de Pater et al., 2009*; *Witzel et al., 2012*) that eventually give rise to the inflow tract (IFT) of the atrium of the mature heart. A second set of SHF cells is positive for Islet family member Islet2b (Isl2b) and for latent TGFβ binding protein 3 (Ltbp3). These cells become the CMs that populate the arterial pole of the heart tube and eventually contribute to the OFT of the ventricle in the mature heart (*Zhou et al., 2011*; *Witzel et al., 2017*).

Given that CMs derive from the FHF and SHF cells of the mesoderm, the final CM number in the mature heart depends in part on the specification of mesodermal cells to form CPCs. Nkx2.5, Mef2c, and Hand2 are known to promote CM differentiation in zebrafish (*Guner-Ataman et al., 2013*; *Hinits et al., 2012*; *Lazic and Scott, 2011*; *Schindler et al., 2014*). Several signaling pathways have been found to act upstream of some of these transcription factors and to restrict the HF at the rostral and caudal boundaries of the ALPM. At the rostral border of the zebrafish ALPM, Tal1 and Etv2, two transcription factors required for vascular and hematopoietic lineage specification, respectively, repress cardiac specification, thereby reducing the number of CMs in the mature heart (*Schoenebeck et al., 2007*). At the border between the zebrafish ALPM and posterior LPM (PLPM), retinoic acid (RA) signaling from the adjacent forelimb field determines the HF size by restricting the potential differentiation of ALPM cells into heart precursor cells, and thus limits the number of atrial, but not ventricular, cells in the mature heart (*Waxman et al., 2008*).

In addition to molecular pathways regulating the fate decisions of CPCs, the final number of CMs is also determined by signalling pathways that regulate cell proliferation at each stage of cardiac morphogenesis (*de Pater et al., 2009*; *Jopling et al., 2010*; *Rochais et al., 2009*; *Vincent and Buckingham, 2010*). One well-characterized signaling pathway is the Hippo signaling pathway, which helps to define the number of cells in a variety of tissues and organs (*Zhou et al., 2015*). In mammalian cells, the active elements of the Hippo pathway include Ste20-like serine/threonine kinase 1 and 2 (Mst1/2, mammalian orthologs of the fruit fly, Hippo), which phosphorylate Large tumour suppressor kinase 1 and 2 (Lats1/2). Phosphorylated Lats1/2 induce the nuclear export of the transcription factor Yes-associated protein 1 (Yap1) and its paralog WW-domain-containing transcription regulator 1 (Wwtr1), also known as Taz. Lats1/2 inhibits the formation of a complex involving Yap1/Wwtr1 and the TEA domain (TEAD) transcription factors by promoting the nuclear export of Yap1/Wwtr1, thereby repressing the expression of downstream target genes (*Zhao et al., 2008*).

In CMs of the mouse heart, Hippo signaling has been implicated in cardiac regeneration after myocardial injury (*Lin et al., 2014*; *von Gise et al., 2012*; *Xin et al., 2013*). While *Yap1* and *Wwtr1* double-null mutations in mice are embryonically lethal before the blastula stage (*Nishioka et al., 2009*), it has been shown that Nuclear Yap1 induces CM proliferation in the adult and fetal mouse. Furthermore, mice that are depleted of *Lats2*, *Salvador* (*Salv*), or *Mst1/2* using CM-specific Cre drivers exhibit a hypertrophic growth due to an increase in CM proliferation (*Zhou et al., 2015*). Together, these results suggest that Hippo signaling plays a key role in cardiac proliferation in the mouse. However, it is unclear whether Yap1/Wwtr1 are involved in CPC proliferation within the FHF and SHF before the formation of the heart tube. In addition, although Hippo signaling also regulates the expression of genes that are essential for cell specification and differentiation (*Zhao et al.,*

*2008*; *Nishioka et al., 2009*), we still do not know whether Hippo signalling plays a role in cardiac cell fate specification.

In the work described here, we sought to examine the role of Hippo signaling in controlling heart cell number beyond its known roles in CM proliferation. Using zebrafish as a model, we examined the role of Hippo signaling at various stages of embryonic development: at the stage when embryos are specifying the HF, at the stage when the heart tube is formed, and in older embryos when heart morphogenesis is largely completed. We demonstrate that Lats1/2-Yap1/Wwtr1-regulated Hippo signaling determines the number of SHF cells in the venous pole that originate from the caudal part of the ALPM. At the molecular level, we show that Yap1/Wwtr1 promote *bone morphogenetic protein-2b* (*bmp2b*) expression and induce the phosphorylation of Smad1/5/9 in both *hand2*- and Isl1-positive cells. Consistently, the absence of *lats1/2* leads to increased *hand2* expression at the boundary between the ALPM and the PLPM and to an increased number of SHF cells in the venous pole. Together, these findings demonstrate that Hippo signaling restricts the number of CPCs located in the venous pole by suppressing Yap1/Wwtr1-dependent Bmp2b expression and *hand2* expression.

## Results

### Lats1/2 are involved in atrial CMs development

To examine whether Yap1/Wwtr1-dependent transcription determines the CM number during early cardiogenesis, we developed *lats1* and *lats2* knockout (KO) fish using transcription activator-like effector nuclease (TALEN) techniques. Fish with *lats1$^{ncv107}$* and *lats2$^{ncv108}$* alleles lack 10 bp at Exon 2 and 16 bp at Exon 3, respectively, resulting in premature stop codons due to frameshift mutations (*Figure 1—figure supplement 1A*). *lats1$^{ncv107}$* KO fish and *lats2$^{ncv108}$* KO fish were viable with no apparent defect (data not shown). However, almost all the *lats1$^{ncv107}$lats2$^{ncv108}$* double KO (*lats1/2* DKO) larvae died before 15 days post-fertilization (dpf) (*Figure 1—figure supplement 1B*).

We assessed the effect of Lats1/2 depletion on heart development by counting CM number in the atrium and the ventricle of *lats1/2* mutant larvae which also contained *Tg(myosin heavy chain 6 [myh6]:Nuclear localization signal [Nls]-tagged tdEosFP);Tg(myosin light polypeptide 7 [myl7]:Nls-mCherry)*. These larvae expressed Nls-tdEosFP in the atrial CMs and Nls-mCherry in the whole CMs (*Figure 1A*). We found that the number of atrial CMs, but not ventricular CMs, was significantly increased in the *lats1$^{wt(wild type)/ ncv107}$lats2$^{ncv108}$* embryos and in the *lats1$^{ncv107}$lats2$^{ncv108}$* embryos at 74 hr post-fertilization (hpf) (*Figure 1B,C* and *Figure 1—source data 1*).

To confirm that Hippo signaling is involved in determining the number of CMs, we assessed whether Yap1/Wwtr1-dependent transcription is activated in developing CMs by analyzing two Tead-reporter transgenic lines: first, we used a general Tead reporter line, the *Tg(eef1a1l1:galdb-hTEAD2ΔN-2A-mCherry)*, which expresses human TEAD2 lacking the amino-terminus (1–113 aa) fused with a GAL4 DNA-binding domain under the control of an *eukaryotic translation elongation factor 1 alpha 1, like 1* (*eef1a1l1*) promoter (*Fukui et al., 2014*); second, we generated a CM-specific Tead reporter line, the *Tg(myl7:galdb-hTEAD2ΔN-2A-mCherry)*, which expresses the same construct under the control of the *myl7* promoter. By crossing these fish with *Tg(uas:GFP)* lines, cells with nuclear-translocated Yap1 or Wwtr1 can be identified through their expression of GFP (*Fukui et al., 2014*).

We first showed that the *lats1/2* double knock-out (DKO) affected Hippo signaling using the general Tead reporter. We found that the Yap1/Wwtr1 reporter was active in the *lats1/2* DKO embryos and in the *lats1/2* morphants (*Figure 1—figure supplement 1C*). We next analyzed the activity of the CM reporter at 74 hpf and found that Yap1/Wwtr1-dependent transcription was present in IFT atrial CMs using the CM Tead reporter line (*Figure 1—figure supplement 2A*).

To identify the stage at which the Tead reporter starts to be active in CMs more precisely, we analyzed the activity of the reporter at an earlier stage. Previous studies have shown that IFT atrial CMs originate from the venous pole of the heart tube (*de Pater et al., 2009*). By examining the progeny of the Tead reporter line crossed with the *Tg(myl7:Nls-mCherry)* line at 24 hpf, we found that the Tead reporter was active in CMs located at the venous pole of the heart tube (*Figure 1—figure supplement 2B*). Importantly, the *eef1a1l1* and 2A peptide-driven mCherry expression used for screening purposes is very weak compared to mCherry expression driven by the *myl7* promoter

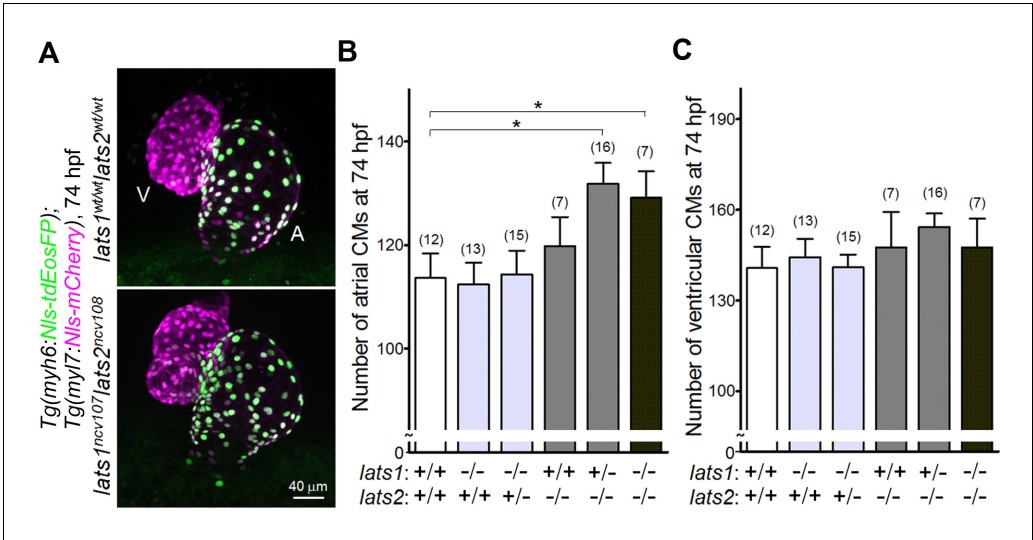

**Figure 1.** Knockout of *lats1/2* genes leads to an increase in the number of atrial, but not ventricular CMs during early development. (**A**) Confocal 3D-stack images (at 74 hr post fertilization [hpf]) of the *Tg(myh6:Nls-tdEosFP);Tg(myl7:Nls-mCherry)* embryos with *lats1^{wt/wt}lats2^{wt/wt}* (top) and *lats1^{ncv107}lats2^{ncv108}* alleles (bottom). Atrial (A) and ventricular (V) cardiomyocytes (CMs) are EosFP-positive cells and EosFP-negative mCherry-positive cells, respectively. Ventral view, anterior to the top. (**B, C**) Quantitative analyses of the number of atrial (**B**) and ventricular (**C**) CMs of the embryos at 74 hpf with alleles indicated at the bottom. Plus (+) and minus (–) signs indicate the *wt* allele and the allele of *ncv107* or *ncv108* in *lats1* or *lats2* genes, respectively. The confocal 3D-stack images are a set of representative images of eight independent experiments. In the graphs, the total number of larvae examined in the experiment is indicated on the top of columns unless otherwise described. $^{*}p < 0.05$.
DOI: https://doi.org/10.7554/eLife.29106.002

The following source data and figure supplements are available for figure 1:

**Source data 1.** Quantification of atrial (*Figure 1B*) and ventricular (*Figure 1C*) cardiomyocyte numbers in the embryos with *lats1* and *lats2* mutant alleles.
DOI: https://doi.org/10.7554/eLife.29106.005

**Figure supplement 1.** Knockout of *lats1/2* genes leads to an activation of the Tead reporter.
DOI: https://doi.org/10.7554/eLife.29106.003

**Figure supplement 2.** Tead reporter expression is found in the venous pole CMs of the atrium.
DOI: https://doi.org/10.7554/eLife.29106.004

and does not affect the intensity analysis. Together, these data suggest that Lats1/2 restrict Yap1/wwtr1-Tead activation during early CM determination.

## Lats1/2 determine the number of IFT CMs derived from Isl1-positive SHF cells

Following on from the observation that the Tead reporter is active at 24 hpf in the progeny of SHF cells, we next assessed whether Hippo signaling affects CM number at this stage. Isl1 is a SHF marker because it plays an essential role in the development of CPCs in the SHF, and because its expression delimitates the SHF as early as 24 hpf in zebrafish (*Caputo et al., 2015*). We generated transgenic fish expressing GFP under the control of an *isl1* BAC promoter; the *TgBAC(isl1:GFP)*. The *isl1* transgene recapitulated the endogenous *isl1* expression patterns (*Figure 2—figure supplement 1A,B*) and expression was found in the IFT of the atrium at four dpf (*Figure 2A,B* and *Figure 2—source data 1*). *isl1*-promoter-active cells were observed in the endocardium and epicardium at 96 hpf (*Figure 2—figure supplement 1C*, arrows and arrowheads), but they were absent from the arterial pole, OFT, and ventricular myocardium until four dpf (*Figure 2A,B* and *Figure 2—source data 1*). To validate our *isl1* reporter further, we made use of Ajuba, a LIM-domain family protein that is known to restrict the number of *isl1*-positive cells in the SHF (*Witzel et al., 2012*). Consistently, *isl1*-promoter-active SHF cells were increased in the *ajuba* morphants (*Figure 2E, Figure 2—figure*

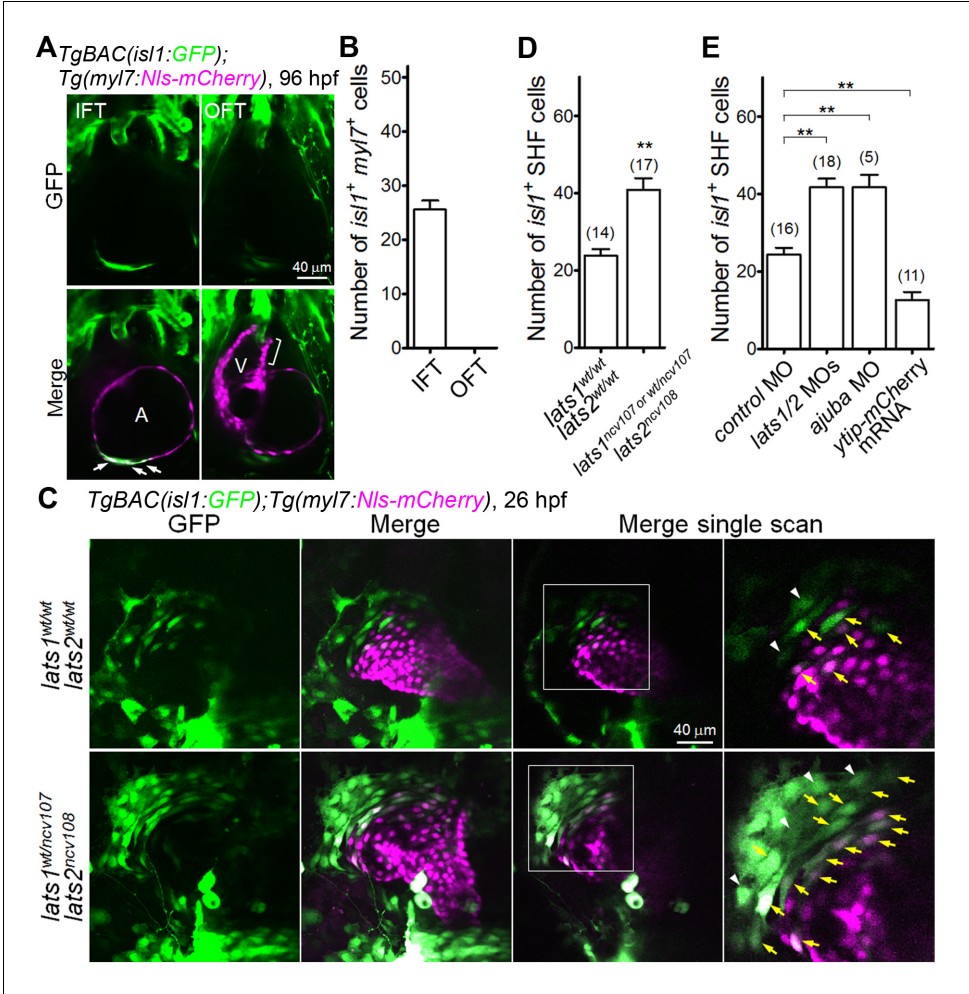

**Figure 2.** The Hippo signaling pathway is involved in the formation of Isl1-positive SHF cells in the venous pole. (A) Single-scan confocal images (at 96 hpf) of the *TgBAC(isl1:GFP);Tg(myl7:Nls-mCherry)* embryos. Cells in which both the *isl1* and *myl7* promoters are active are present in the inflow tract (IFT) cells (arrows) but not in the outflow tract (OFT) cells (square bracket). A, atrium; V, ventricle. Ventral view, anterior to the top. (B) Quantitative analyses of the number of cells with both *isl1*- and *myl7*-promoter activities in the IFT and the OFT at 96 hpf (n = 10). (C) Confocal images (at 26 hpf) of the *TgBAC(isl1:GFP);Tg(myl7:Nls-mCherry)* embryos with the *lats1^{wt/wt}lats2^{wt/wt}* (upper panels) or *lats1^{wt/ncv107}lats2^{ncv108}* alleles (bottom panels). The boxed regions are enlarged in the panels in the fourth (right) column. Yellow arrows indicate both *isl1*- and *myl7*-promoter activities in cells in the venous pole. White arrowheads indicate cells with *isl1*-promoter activity that are in contact with cells in which there is *myl7*-promoter activity. Confocal 3D-stack images (left two panels) and single-scan images (right two panels). Dorsal view, anterior to the top. (D, E) Quantitative analyses of the number of the *isl1*-promoter-active SHF cells in the venous pole of the *lats1^{wt/wt}lats2^{wt/wt}* embryos, the *lats1^{wt/ncv107}lats2^{ncv108}* embryos, and the *lats1/2* DKO embryos (D), and in embryos shown in *Figure 2—figure supplement 1D* (E). Both cells with *isl1*- and *myl7*-promoter activity and with *isl1*-promoter activity that were in contact cells with *myl7*-promoter activity were counted as SHF cells. The confocal 3D-stack images and single-scan (2 µm) images are a set of representative images of at least four independent experiments. **p < 0.01.

DOI: https://doi.org/10.7554/eLife.29106.006

The following source data and figure supplements are available for figure 2:

**Source data 1.** The number of *isl1*- and *myl7*-promoter-active cells in IFT and OFT cells at 96 hpf (*Figure 2B*), the numbers of *isl1*-promoter-active SHF cells in *lats1/2* mutants (*Figure 2D*) and the number of *isl1*-promoter-active SHF cells of embryos at 26 hpf injected with morpholino (MO) and mRNA (*Figure 2E*).
DOI: https://doi.org/10.7554/eLife.29106.009

**Figure supplement 1.** GFP expression in *isl1* BAC transgenic fish recapitulates endogenous *isl1* expression.
DOI: https://doi.org/10.7554/eLife.29106.007

*Figure 2 continued on next page*

*Figure 2 continued*

**Figure supplement 2.** Depletion of Lats1/2 leads to an increase in the number of both Tead-reporter and Isl1-positive SHF cells in the venous pole.

DOI: https://doi.org/10.7554/eLife.29106.008

*supplement 1D* and *Figure 2—source data 1*). Together, these results suggest that the *TgBAC(isl1: GFP)* line recapitulates *isl1* expression in vivo.

We found that a significant number of Tead reporter-positive CMs were positive for Isl1 in the venous pole (*Figure 2—figure supplement 2A*). Furthermore, the number of both Isl1- and Tead-reporter-positive CMs in the venous pole was significantly increased in the *lats1/2* morphants (*Figure 2—figure supplement 2B*). Making use of the *TgBAC(isl1:GFP)* line, we next quantified the population of *isl1*-promoter-positive SHF cells in the *lats1/2* DKO embryos at 26 hpf. As expected, the number of *isl1*-promoter-active cells in the venous pole was significantly increased in both *lats1^{wt/ ncv107}lats2^{ncv108}* and *lats1/2* DKO embryos (*Figure 2C,D* and *Figure 2—source data 1*). Consistent with this, the *isl1*-promoter-active SHF cells were significantly increased in the venous pole of the *lats1/2* morphants (*Figure 2E*, *Figure 2—figure supplement 1D* and *Figure 2—source data 1*). To further confirm the importance of Hippo signaling in the SHF, we analyzed embryos expressing a mCherry-tagged dominant-negative form of Yap1/Wwtr1-Tead-dependent transcription (ytip-mCherry) (*Fukui et al., 2014*) and found that the number of *isl1*-positive cells were significantly decreased in the venous pole (*Figure 2E*, *Figure 2—figure supplement 1D* and *Figure 2—source data 1*). Together, these results demonstrate that Lats1/2-mediated Hippo signaling is involved in reducing the number of SHF-derived CPCs that contribute to the venous pole.

## Lats1/2 determine the number of CMs derived from the *hand2*-promoter-active CMs

To identify the mechanism of action of Hippo signaling, we sought to identify the gene targets of Yap1/Wwtr1 in early CPC differentiation. We examined whether Yap1/Wwtr1 regulate the expression of transcription factors *nkx2.5*, *hand2*, and *gata4*, all of which are essential for early CPC differentiation (*Schoenebeck et al., 2007*). qPCR revealed that *hand2* mRNA expression was significantly upregulated (*Figure 3A* and *Figure 3—source data 1*) and that *nkx2.5* and *gata4* mRNAs expression was unaffected in the *lats1/2* morphants (*Figure 3A* and *Figure 3—source data 1*). Whole-mount in situ hybridization (WISH) analyses revealed that the *hand2* expression domain, corresponding to the region that gives rise to the heart, was expanded in the *lats1^{wt/ ncv107}lats2^{ncv108}* embryos, the *lats1/2* DKO embryos, and the *lats1/2* morphants at 22 hpf (*Figure 3B* and *Figure 3—figure supplement 1A*). These data suggest that Lats1/2 determine the number of atrial CMs by inhibiting Yap1/Wwtr1-dependent *hand2* expression.

Overexpression of Hand2 increases the number of SHF-derived CMs in zebrafish (*Schindler et al., 2014*), so we hypothesized that the increased number of CM in *lats1/2* mutants is due to an increase in the number of CPCs in the SHF. We first investigated endogenous *hand2* expression during CM development by analyzing *TgBAC(hand2:GFP)* (*Yin et al., 2010*), which labels cells in which the *hand2* promoter is activated with GFP, and *Tg(myl7:Nls-mCherry)*, which labels CMs and CM progenitors with nuclear-localized mCherry. At 26 hpf, we found that *hand2*-promoter-active CMs are localized on the anterior side of the growing cardiac tube, corresponding to the venous pole (*Figure 3C*). In the *lats1/2* DKO embryos, the number of *hand2*-promoter-active CMs was significantly increased in the venous pole (*Figure 3C,D* and *Figure 3—source data 1*). Similarly, the number of *hand2*-promoter-active CMs was increased in the venous pole in the *lats1/2* morphants (*Figure 3—figure supplement 1B*). Furthermore, we found that the population of Isl1-positive SHF cells overlaps with the *hand2*-promoter-active CMs in the very left-rostral end of the cardiac tube (*Figure 3E*, brackets). As expected, the population of Isl1-positive SHF cells was increased in the *lats1/2* DKO embryos and in the *lats1/2* morphants (*Figure 3E* and *Figure 3—figure supplement 1C*, brackets). Thus, *hand2* is expressed in differentiated CMs and the *hand2* expression domain contains SHF cells at 26 hpf.

To confirm the involvement of the Hippo pathway in modulating the *hand2* expression domain, we next sought to examine the number of *hand2-promoter*-active CMs in the DKO mutants of *yap1* (*Figure 3—figure supplement 2A*) and *wwtr1* (*Nakajima et al., 2017*). As expected, we found that

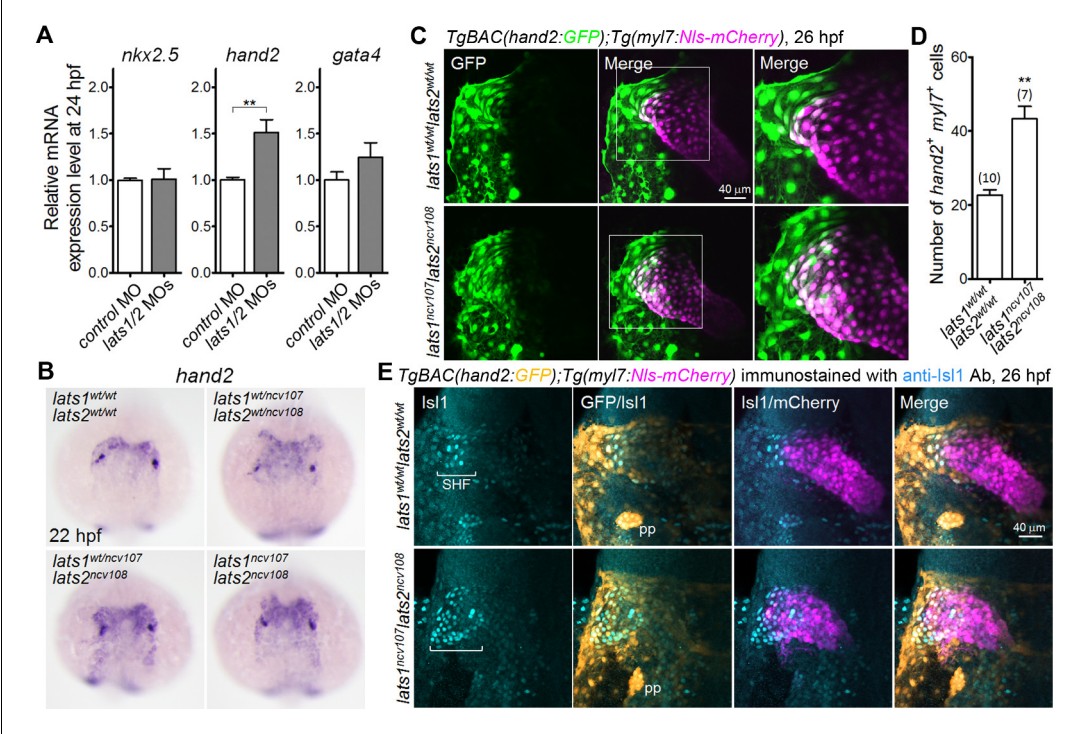

**Figure 3.** Knockout of *lats1/2* results in an increase in the number of cells in the venous pole in which both *myl7* and *hand2* promoters are activated. (**A**) Quantitative-PCR analyses of the expression of *nkx2.5*, *hand2*, and *gata4* mRNAs in the whole embryos at 24 hpf showing the effects of MO injection (n = 4). Relative expression of mRNA in the MO-injected morphants to that of the control is shown. (**B**) WISH analyses at 22 hpf of *lats1^{wt/wt}lats2^{wt/wt}* (n = 7), *lats1^{wt/ncv107}lats2^{wt/ncv108}* (n = 22), *lats1^{wt/ncv107}lats2^{ncv108}* (n = 14) and *lats1^{ncv107}lats2^{ncv108}* (n = 6) embryos using an antisense probe for *hand2*. (**C**) Confocal 3D-stack images (at 26 hpf) of *TgBAC(hand2:GFP);Tg(myl7:Nls-mCherry)*-labeled embryos carrying the *lats1^{wt/wt}lats2^{wt/wt}* (upper panels) or *lats1^{ncv107}lats2^{ncv108}* allele (bottom panels). GFP images (left), merged GFP and mCherry images (center), and enlarged images of the boxed regions in the center panels (right) are shown. (**D**) Quantitative analysis of the number of cells in which both *hand2* and *myl7* promoters are activated at 26 hpf. (**E**) Confocal 3D-stack images (at 26 hpf) of the *TgBAC(hand2:GFP);Tg(myl7:Nls-mCherry)*-labeled embryos containing the *lats1^{wt/wt}lats2^{wt/wt}* (upper panels, n = 9) or the *lats1^{ncv107}lats2^{ncv108}* allele (bottom panels, n = 5) immunostained with the anti-Isl1 antibody (anti-Isl1 Ab). Square brackets denote the SHF cells that are Isl1-positive, both *hand2-* and *myl7*-promoter-active cells that are Isl1-positive, and *hand2*-promoter-active cells that are in contact with *myl7*-promoter-active cells. pp indicates the pharyngeal pouch, which expresses the *hand2*-promoter-activated GFP signal. The first, second, third and fourth columns show Isl1 immunostaining, merged images of GFP and Isl1 immunostaining, merged images of Isl1 immunostaining and mCherry labeling, and merged images of all the three (GFP, mCherry, and Isl1 immunostaining), respectively. All of the images are of the dorsal view, anterior to the top. The confocal 3D-stack images and the WISH images are a set of representative images from at least four independent experiments. ^{**}p < 0.01.
DOI: https://doi.org/10.7554/eLife.29106.010

The following source data and figure supplements are available for figure 3:

**Source data 1.** Quantification of the relative mRNAs expression levels (**Figure 3A**) and the numbers of both *hand2-* and *myl7*-promoter-active cells (**Figure 3D**).
DOI: https://doi.org/10.7554/eLife.29106.013

**Figure supplement 1.** Depletion of Lats1/2 results in an increase in the number of *hand2*-promoter-active cells in the venous pole.
DOI: https://doi.org/10.7554/eLife.29106.011

**Figure supplement 2.** *hand2*-promoter-active cells were significantly decreased in *yap1/wwtr1* double-knockout embryos.
DOI: https://doi.org/10.7554/eLife.29106.012

the extension of the embryo along the anterior-posterior axis is severely impaired in *yap1* and *wwtr1* DKO embryos (**Figure 3—figure supplement 2B**) (*Kimelman et al., 2017*; *Nakajima et al., 2017*). When analyzing the *hand2*-promoter-active cells, we found that their number is greatly reduced (**Figure 3—figure supplement 2C**). Interestingly, the DKO mutant embryos exhibit cardia bifida, which is also observed in the *hand2* mutant (*Yelon et al., 2000*) (**Figure 3—figure supplement 2C,D**). Together, these results strongly suggest that the increased CPC number in the *lats1/2* mutants is due to an expansion of the *hand2* expression domain in the SHF.

## hand2-promoter-active cells at the caudal end of the ALPM migrate toward the venous pole of the cardiac tube

In zebrafish, the origin of venous pole CMs is unknown. In amniotes, the venous pole progenitors are located in the most caudal domain of the ALPM (*Abu-Issa and Kirby, 2008*; *Galli et al., 2008*). Considering that *hand2* is expressed in the zebrafish LPM (*Schoenebeck et al., 2007*), we hypothesized that *hand2*-promoter-active cells originate from the caudal end of the ALPM. We performed time-lapse imaging from 14 hpf to 26 hpf to investigate whether *hand2*-promoter-active cells of the LPM contribute to the venous pole cells. Cell-tracking analysis revealed that the caudal cells of the ALPM migrate toward the venous pole (*Figure 4A,B*, and *Video 1*). ALPM cells move toward the posterior of the cardiac disc by 20 hpf, and subsequently move anterior-laterally toward the venous pole of the cardiac tube by 26 hpf (*Figure 4A–C*). These results indicate that *hand2*-promoter-active cells in the venous pole differentiate from the caudal ALPM. To confirm that the caudal ALPM cells are incorporated into the IFT atrial CMs, we sought to perform cell-tracking of *hand2*-promoter-active cells in the caudal region of ALPM cells following photoconversion. To do so, we injected embryos with a plasmid that expresses tdEosFP under the control of *hand2* BAC promoter and photoconverted the cells in the caudal region of both sides of ALPM in these embryos. We found that the photoconverted cells were incorporated into the IFT of the atrium and the OFT of the ventricle (*Figure 4D*). This indicates that the cells of the caudal region of both sides of ALPM can become IFT CMs of the atrium.

We next analyzed whether the Tead reporter is active in the *hand2* expression domain of the ALPM. We crossed *TgBAC(hand2:GFP)* lines with general Tead mRFP1 reporter fish. At the 12 somite stage (ss), the Tead-reporter-active cells were found in the entire region of the ALPM and overlapped with *hand2*-promoter-active cells in ALPM (*Figure 4E*). We further noticed that the Tead reporter was inactive in the rostral region of the PLPM at 12 ss (*Figure 4—figure supplement 1A*). These data suggest that Hippo signaling acts upstream of *hand2* expression in the ALPM and may play a role in determining CM fate in the ALPM.

## Hippo signaling regulates the number of SHF cells from the caudal end of the ALPM

Tead reporter activation in the cells of ALPM prompted us to ask whether Lats1/2-Yap1/Wwtr1 signaling is involved in the proliferation and/or specification of those cells. We examined the proliferation of *isl1*-promoter-active cells using the EdU incorporation assay and found that the number of *isl1*-promoter-active and EdU-positive CMs in the *lats1/2* morphants was comparable to that of controls (*Figure 5A,B*). In addition, there was no difference in the number of EdU-positive blood cells and endocardial cells among the two groups (data not shown). Importantly, the timing of EdU incorporation did not affect the results of the proliferation analyses (*Figure 5B*), suggesting that the increase in the number of *isl1*-promoter-active SHF cells that resulted from the depletion of Lats1/2 is not caused by cell proliferation after the differentiation of SHF cells from the ALPM.

We next tested whether Lats1/2 affect SHF cell specification in the ALPM. At 10 ss, the ALPM and the PLPM can be characterized by the expression of *gata4*, *nkx2.5*, *tal1*, and *hand2*. *gata4* labels the multipotent myocardial-endothelial-myeloid progenitors of the ALPM; *nkx2.5* is a marker for the ventricular HF; *tal1* marks the hematopoietic cell progenitors; *hand2* marks both the ALPM and the PLPM at 10 ss (*Figure 5C*) with a clear gap in between. Interestingly, the gap length of the *lats1*$^{wt/ncv107}$*lats2*$^{ncv108}$ embryos, the *lats1/2* DKO embryos, and the *lats1/2* morphants was significantly shorter than that of wildtype embryos (*Figure 5D,E*, *Figure 5—figure supplement 1A* and *Figure 5—source data 1*). We also found that *tal1* expression was decreased in the rostral end of the PLPM in the *lats1/2* DKO embryos and the *lats1/2* morphants (*Figure 5F* and *Figure 5—figure supplement 1B*). A similar analysis in *yap1/wwtr1* DKO embryos revealed that *hand2* expression was decreased in both ALPM and PLPM but that the expression of *tal1* was unaffected (*Figure 5D,F*). Importantly, the expression of *gata4* and *nkx2.5* was unaffected in *lats1/2* DKO embryos and in *lats1/2* morphants, suggesting that Hippo signaling mainly affects the *hand2* expression domain (*Figure 5G*, and *Figure 5—figure supplement 1C*). These observations were consistent with the results of qRT-PCR (*Figure 3A*).

To confirm the specificity of Hippo action, we examined the expression of *etv2*, which is a marker for blood-vessel progenitors (*Schoenebeck et al., 2007*), and of *hoxb5b*, which is a regulatory

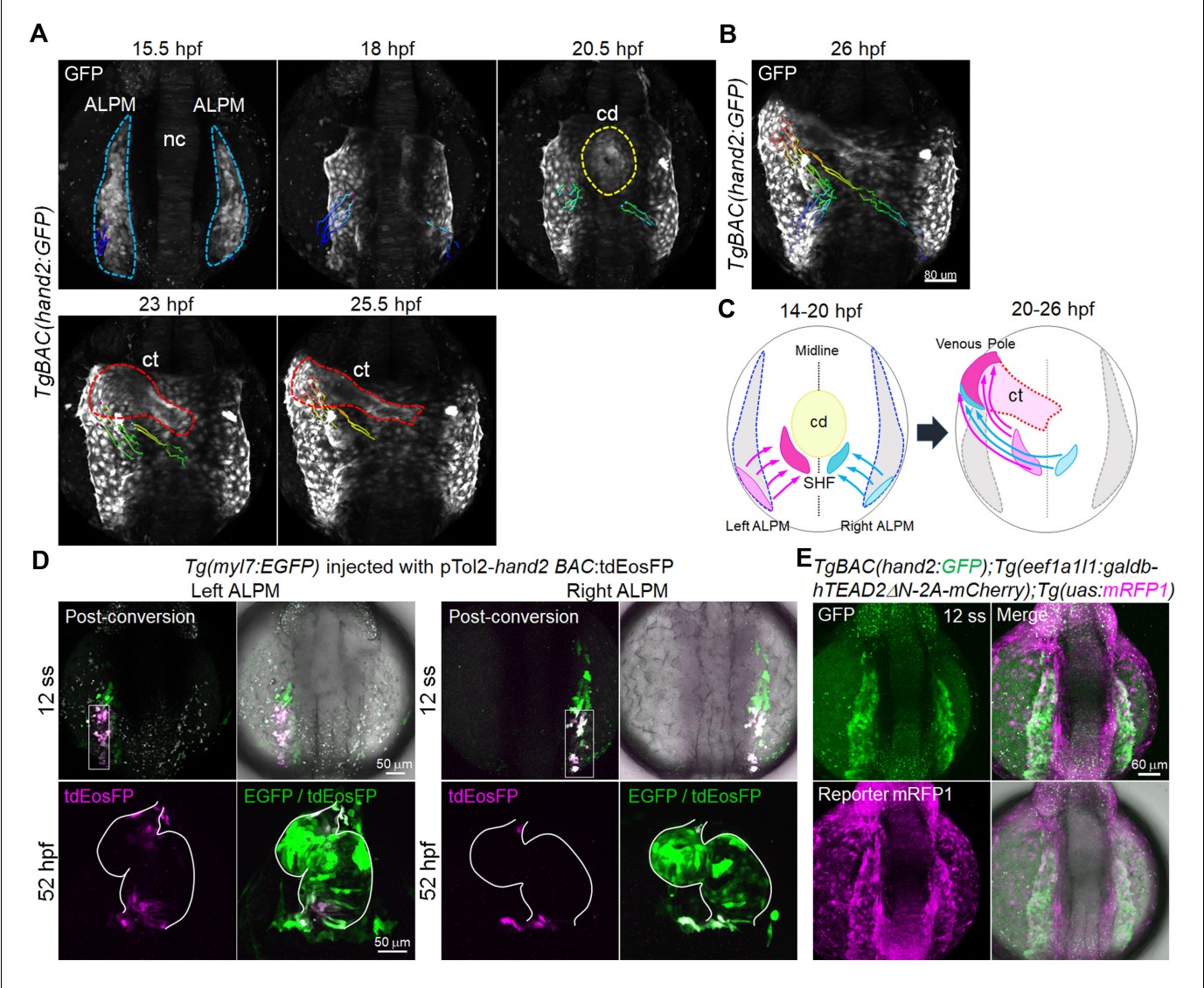

**Figure 4.** Tead-reporter-active cells in the caudal region of the ALPM move to the venous pole. (A) Time-sequential 3D-rendered confocal images of the *TgBAC(hand2:GFP)* embryo from 15.5 hpf to 25.5 hpf (n = 6). Spots of magenta and cyan denote the cells in the caudal part of the left and right ALPM, respectively. Notochord, nc; cardiac disc, cd; cardiac tube, ct. ALPM, cd, and ct are indicated by the blue, yellow, and red broken lines, respectively. (B) Tracking of caudal end *hand2*-promoter-active ALPM cells from 14 hpf to 26 hpf. The 3D-rendered confocal image of the *TgBAC (hand2:GFP)* embryo at 26 hpf with the track of cells showing color changes from blue to red according to the time after imaging (14 hpf to 26 hpf). (C) Schematic illustration of the trajectory patterns of the caudal end ALPM cells from 14 hpf to 26 hpf. Magenta and cyan denote the caudal region of the left and right ALPM, respectively. The cells in the caudal region of the left ALPM (magenta) and the right ALPM (cyan) moved to the venous pole (26 hpf) through the SHF region posterior to the cd (20 hpf). (D) Confocal 3D-stack images of the *Tg(myl7:EGFP)* embryo injected with pTol2-*hand2* BAC: tdEosFP at the 12 somite stage (ss) (upper panels) and at 52 hpf (lower panels). Cells from the caudal region of either left ALPM (left four panels, n = 19) or right ALPM (right four panels, n = 18) were photoconverted at 12 ss. The hearts of the photoconverted embryos were imaged at 52 hpf. White squares and white lines indicate the photoconverted area and the outline of the heart, respectively. (E) Confocal 3D-stack images (at 12 ss) of the *TgBAC(hand2:GFP);Tg(eef1a1l1:galdb-hTEAD2ΔN-2A-mCherry);Tg(uas:mRFP1)* embryo (n = 6). Tead-reporter-active cells were present in the entire ALPM. Images of the ALPM are in dorsal view, anterior to the top. Images of the heart are in ventral view, anterior to the top. The confocal 3D-stack images are a set of representative images from at least five independent experiments.

DOI: https://doi.org/10.7554/eLife.29106.014

The following figure supplement is available for figure 4:

**Figure supplement 1.** The Tead reporter is expressed in the ALPM but not in the rostral region of the PLPM.

DOI: https://doi.org/10.7554/eLife.29106.015

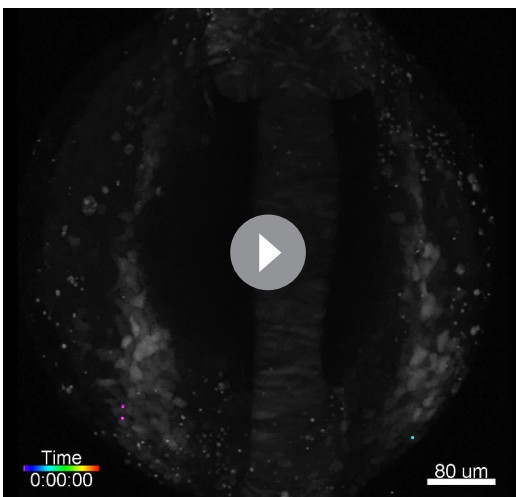

**Video 1.** *hand2*-promoter-active cells in the caudal region of the ALPM move to the venous pole. Time-lapse recording of 3D-rendered confocal images of the *TgBAC*(*hand2:GFP*) embryo from 14 hpf (10 ss) to 26 hpf. Note the migration of the caudal region of both the left ALPM (magenta) and the right ALPM (cyan) toward the venous pole of the cardiac tube. Changes in the colors reflect the tracking time (blue, 0 hr; red, 12 hr). Dorsal view, anterior to the top. The time-lapse movie is a set of representative data from six independent experiments. *Video 1* is related to *Figure 4A,B*.
DOI: https://doi.org/10.7554/eLife.29106.016

molecule of RA signaling in the forelimb field (*Waxman et al., 2008*) in the LPM. The expression levels of both *etv2* and *hoxb5b* were comparable between the control and the *lats1/2* morphants (*Figure 5—figure supplement 1C*). Collectively, these results suggest that Lats1/2 negatively regulate Yap1/Wwtr1-dependent differentiation of the LPM into the SHF at the boundary between ALPM and PLPM.

## Hippo signaling regulates Bmp-dependent smad activation that determines the number of SHF cells in the venous pole

Signaling mediated by Bone morphogenetic proteins (Bmps) affects various contexts of heart development via Smad phosphorylation-dependent transcriptional activation. Bmp-Smad signaling is known to be essential for SHF formation, FHF-derived CM development, endocardium development and epicardium development (*Prall et al., 2007*; *Schlueter et al., 2006*; *Tirosh-Finkel et al., 2010*; *Yang et al., 2006*). Yap1 is known to promote *Bmp2b* expression in zebrafish neocortical astrocyte differentiation (*Huang et al., 2016*) and Bmp2 in mouse endothelial cells (*Neto et al., 2018*). In the zebrafish embryo, *bmp2b*, but not *bmp4*, is expressed in the LPM (*Chung et al., 2008*). We hypothesized that Yap1/Wwtr1 are involved in *bmp2b*-dependent signaling during early cardiogenesis. To investigate whether Bmp-Smad signaling is acti-

vated in the ALPM, we examined Bmp-dependent transcription using the *Tg(BRE:GFP)* fish embryos, in which the Bmp-responsive element (BRE) drives GFP expression (*Collery and Link, 2011*). At 14 hpf, BRE-positive cells were found in the ALPM (*Figure 6—figure supplement 1A*). Because Bmps induce the phosphorylation of Smad1/5/9 (Smad9 is also known as Smad8) (*Heldin et al., 1997*), we examined the phosphorylation of Smad1/5/9 in the embryos at 14 hpf using immunohistochemistry. The phosphorylated Smad1/5/9-positive cells were found in the ALPM and eyes (*Figure 6A*). Phosphorylation of Smad1/5/9 was enhanced in the *lats1/2* DKO embryos and decreased in the *yap1/wwtr1* DKO embryos (*Figure 6A*). At 10 ss, *bmp2b* expression was increased in the ALPM and eyes of the *lats1/2* DKO embryos and decreased in the *yap1/wwtr1* DKO embryos (*Figure 6B*). Consistently, *bmp2b* mRNA was increased in the *lats1/2* morphants at 10 ss (*Figure 6—figure supplement 1B*). Although we could not detect *bmp4* in the ALPM in the early ss (data not shown), *bmp4* mRNA was increased in the venous pole of the *lats1/2* morphants at 26 hpf (*Figure 6—figure supplement 1C*). These results suggest that Hippo signaling functions upstream of Bmp-dependent Smad activation in the ALPM during early cardiogenesis.

By analyzing the BRE reporter, we found that the number of Bmp signal-active cells marked by GFP in the venous pole was increased in the *lats1$^{wt/ncv107}$lats2$^{ncv108}$* embryos and/or the *lats1/2* DKO embryos, as well as in the *lats1/2* morphants at 24 hpf (*Figure 6C,D*, *Figure 6—figure supplement 2A* and *Figure 6—source data 1*). Immunohistochemistry revealed that the number of phosphorylated Smad1/5/9-positive and *hand2*-promoter-active cells was also increased at the venous pole of the *lats1/2* morphants at 26 hpf (*Figure 6—figure supplement 2B*). These results suggest that Yap1/Wwtr1 promote *bmp2b* expression and subsequent Bmp signaling and that Lats1/2 restrict Yap1/Wwtr1-dependent Bmp signaling leading to the formation of the proper venous pole.

We further investigated whether Bmp-Smad activation promotes *hand2* expression. We made use of Smad7, an inhibitory-Smad that blocks Bmp-Smad signaling by interacting with activated Bmp

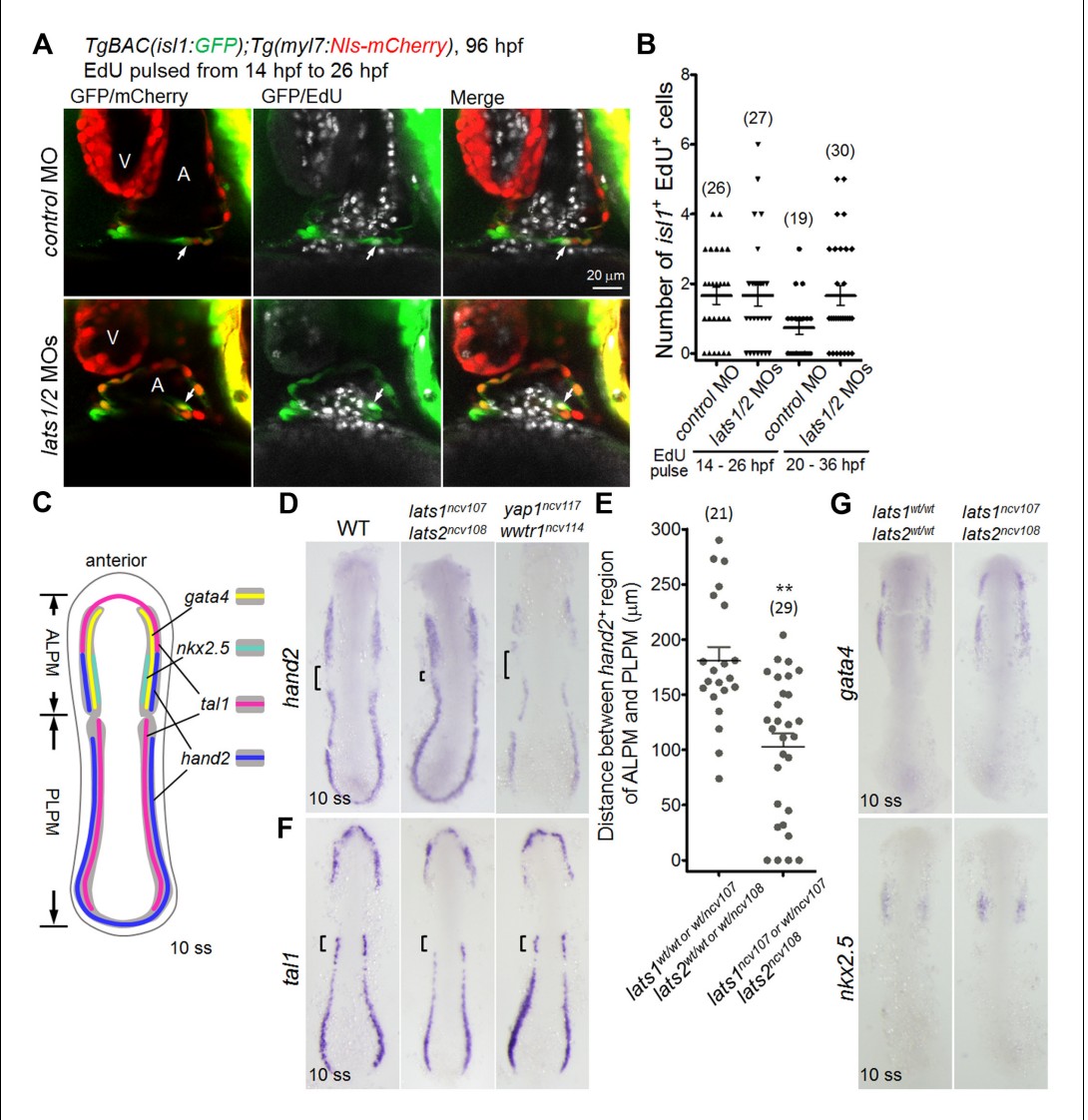

**Figure 5.** Knockout of *lats1/2* leads to an increase in the expression of *hand2* in the boundary between ALPM and PLPM. (**A**) Single-scan confocal images (at 96 hpf) of the *TgBAC(isl1:GFP);Tg(myl7:Nls-mCherry)* embryos injected with MO and pulsed with EdU from 14 hpf to 26 hpf. Arrows indicate the EdU-incorporated *isl1-* and *myl7-*promoter-active cells in the IFT of the atrium. A, atrium; V, ventricle. Ventral view, anterior to the top. (**B**) The number of EdU-positive *isl1-*promoter-active CMs among the embryos treated with MO. Embryos pulsed with EdU from 14 hpf to 26 hpf (left two columns) and from 20 hpf to 36 hpf (right two columns). (**C**) Schematic illustration of gene expression patterns in the LPM of wildtype (WT) embryos at 10 somite stage (ss). Expression domain of *tal1*, *gata4*, *nkx2.5* and *hand2* are depicted as magenta, yellow, green, and blue, respectively. Dorsal view, anterior to the top. (**D, F, G**) WISH analyses of the embryos at 10 ss using the antisense probes indicated to the left of the panels. (**D, F**) Genotypes are WT (left panels, n = 8 to 18), *lats1/2* DKO (center panels, n = 6 to 13), and *yap1/wwtr1* DKO (right panels, n = 5 to 7). (**D**) Square brackets indicate the gap between *hand2*-positive regions of ALPM and PLPM. (**E**) Quantitative measurement of the distance indicated by the brackets in (**D**) in either the *lats1^{wt/wt}lats2^{wt/wt}* embryos or the *lats1^{wt/ncv107}lats2^{wt/ncv108}* embryos and in either the *lats1^{wt/ncv107}lats2^{ncv108}* or the *lats1/2* DKO embryos. (**F**) Brackets indicate the *tal1*-positive rostral end of PLPM in the WT. (**G**) Genotypes are WT (left panels, n = 4 to 5) and *lats1/2* DKO (right panels, n = 3 to 4). Dorsal view, anterior to the top. The single-scan (2 μm) confocal images and in situ hybridization (ISH) images are a set of representative images of at least four independent experiments. **p < 0.01.

DOI: https://doi.org/10.7554/eLife.29106.017

The following source data and figure supplement are available for figure 5:

*Figure 5 continued on next page*

*Figure 5 continued*

**Source data 1.** Distance between *hand2*-positive regions of ALPM and PLPM in the control and in the *lats1/2* mutants at 10 ss (*Figure 5E*).

DOI: https://doi.org/10.7554/eLife.29106.019

**Figure supplement 1.** Depletion of Lats1/2 leads to an increase in the expression of *hand2*.

DOI: https://doi.org/10.7554/eLife.29106.018

type I receptors and preventing the downstream activation of receptor-regulated Smads (*Souchelnytskyi et al., 1998*). We first showed that overexpression of *smad7* mRNA caused dorsalization, demonstrating that Smad7 is essential for proper Bmp-Smad signaling, using 200 pg of mRNA (*Figure 6—figure supplement 3A,B*). Interestingly, the injection of lower concentration of mRNA (100 pg) led to a dorsalization phenotype in only 10% of the injected embryos. Therefore, the remaining embryos without dorsalization phenotype were used to assess heart development (*Figure 6—figure supplement 3A,B*). The non-dorsalized embryos looked healthy. The injection of *smad7* mRNA did not cause fragmentation of the cells (data not shown), but the embryos exhibited a decreased number of *isl1*-promoter-active cells in the *TgBAC(isl1:GFP);Tg(myl7:Nls-mCherry)* assay (*Figure 6E,F* and *Figure 6—source data 1*). Collectively, these results suggest that Bmp-dependent signaling is required for CPC fate determination.

Finally, to confirm the necessity of Bmp-Smad-regulated signaling during SHF formation, we treated the *TgBAC(hand2:GFP);Tg(myl7:Nls-mCherry)* embryos and the *TgBAC(isl1:GFP);Tg(myl7:Nls-mCherry)* embryos with a Bmp inhibitor, DMH1, from 14 hpf to 26 hpf. The efficiency of DMH1 was confirmed by decreased phosphorylation of Smad1/5/9 (*Figure 6—figure supplement 2C*). The expression of Isl1 and the promoter activity of *hand2* were greatly reduced in the embryos treated with DMH1 (*Figure 6—figure supplement 3C*). The number of *isl1*-promoter-active SHF cells was decreased in the embryos treated with DMH1, whereas the number of *isl1*-promoter-inactive CMs in the DMH1-treated embryos was comparable to that in the control embryos (*Figure 6G,H*, and *Figure 6—source data 1*). We thus conclude that Lats1/2 restrict Yap1/Wwtr1-promoted Bmp2b-dependent signaling, which is required for both *hand2*- and *isl1*-promoter activity during SHF formation.

## Discussion

Here, we show for the first time that the Hippo signaling pathway is involved in the determination of LPM cell fate by promoting venous pole identity and increases in atrial CM number (*Figure 7*). We show that Yap1/Wwtr1-promoted signaling increases the size of the SHF domain and that Lats1/2, by inhibiting Yap1/Wwtr1 activity, restrict it. We propose that the increased number of SHF cells in the *lats1/2* DKO embryos may result from a change in fate determination of *hand2*-negative cells, which become *hand2*-positive cells in the boundary between ALPM and PLPM. Indeed, we found that the expression of the marker of blood-cell progenitors *tal1* was repressed in the rostral region of PLPM in *lats1/2* DKO embryos. *lats1/2* mutants exhibited a subtle increase in the number of Isl1-positive atrial SHF cells, with no defect apparent in other organs. Importantly, despite the known role of Hippo signaling in CM proliferation during heart maturation, Hippo signaling was not found to affect cell proliferation during these early stages. Therefore, Hippo signaling contributes specifically to the determination of LPM differentiation.

We speculate that Hippo signaling cooperates with other signaling pathways to determine HF formation. For example, *hoxb5b* expression in the forelimb field limits the extent of the HF at the posterior border of the ALPM, cells of which differentiate into atrial but not ventricular CMs (*Waxman et al., 2008*). Furthermore, other signals generated in the pronephric field in the intermediate mesoderm and in the angiogenic field in the rostral region of PLPM are important for the regulation of cell fate at the posterior HF boundary (*Kimmel et al., 1990*; *Mudumana et al., 2008*).

Our results also help to clarify the origin of venous pole CMs in zebrafish and its link to the SHF. In the mouse embryo, the anterior and posterior SHF cells differentiate into the OFT/right ventricular myocardium and the IFT/atrial myocardium, respectively (*Galli et al., 2008*; *Verzi et al., 2005*). The posterior-SHF in the HF is located caudally (*Abu-Issa and Kirby, 2008*; *Galli et al., 2008*). Tead-reporter activation occurred in the caudal zone of the ALPM, and these cells were shown to become

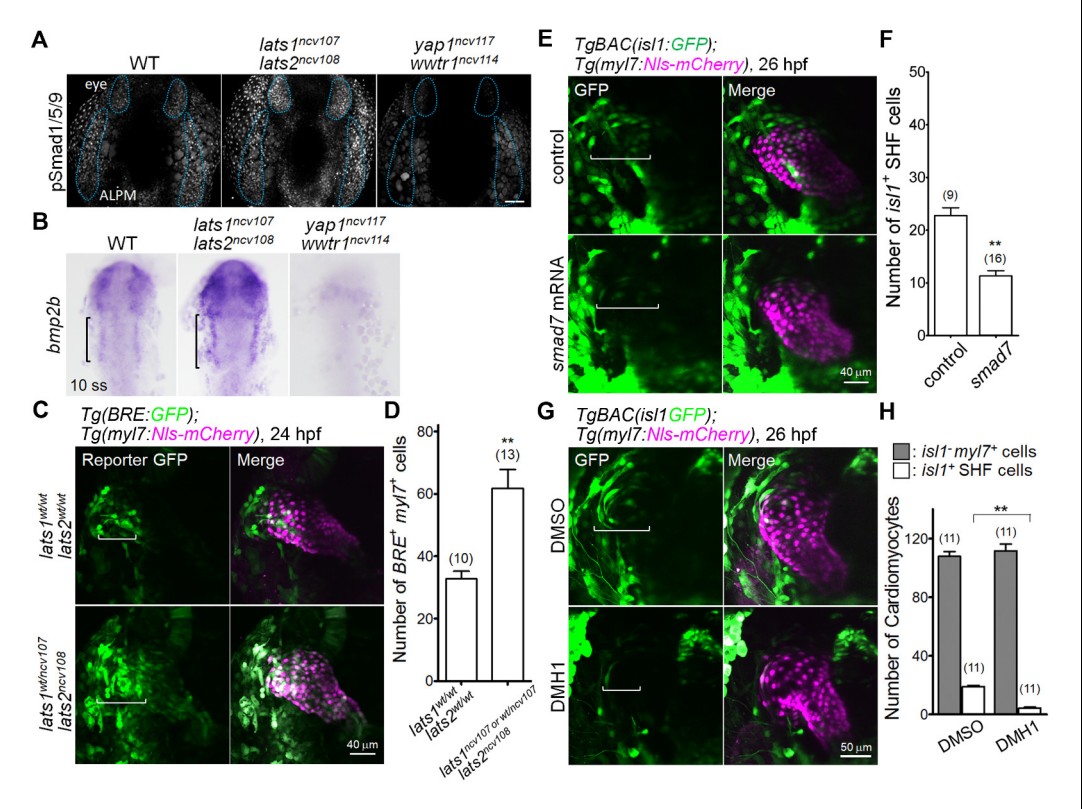

**Figure 6.** The Hippo signaling pathway functions upstream of the Bmp-dependent signal that is necessary for Isl1-positive SHF formation. (A) Confocal 3D-stack images (at 14 hpf) of embryos immunostained with the anti-pSmad1/5/9 Ab. Blue broken lines indicate the phosphorylated Smad1/5/9-positive cells in the ALPM and eyes. Note that the pSmad1/5/9-positive signals in the ALPM and the eyes are increased and decreased in the embryos of $lats1/2$ DKO and $yap1/wwtr1$ DKO, respectively (n = 3). Scale bar indicates 60 μm. (B) WISH analyses at the 10 somite stage (ss) of WT (n = 9), $lats1/2$ DKO (n = 10), and $yap1/wwtr1$ DKO (n = 8) embryos, using antisense probe for $bmp2b$. Square brackets indicate the $bmp2b$-positive ALPM. (C) Confocal 3D-stack images (at 24 hpf) of the $Tg(BRE:GFP);Tg(myl7:Nls-mCherry)$ embryos carrying $lats1^{wt/wt}lats2^{wt/wt}$ (upper panels) and $lats1^{wt/ncv107}lats2^{ncv108}$ alleles (bottom panels). Square brackets highlight the GFP-positive $myl7$-promoter-active cells in the venous pole. Note that the numbers of GFP-positive $myl7$-promoter-active cells are increased in the venous pole. (D) Quantitative analyses of the numbers of the both $BRE$-active GFP-positive and $myl7$-promoter-active mCherry-positive cells in the $lats1^{wt/wt}lats2^{wt/wt}$ embryos and in either the $lats1^{wt/ncv107}lats2^{ncv108}$ embryos or the $lats1/2$ DKO embryos. (E) Confocal 3D-stack images (at 26 hpf) of the $TgBAC(isl1:GFP);Tg(myl7:Nls-mCherry)$ control embryos (uninjected, upper panels) and embryos injected with 100 pg of $smad7$ mRNA (bottom panels). Square brackets indicate the region of $isl1$-promoter-active GFP-positive SHF cells in the venous pole. (F) Quantitative analyses of the number of $isl1$-promoter-active SHF cells in the venous pole. Note that overexpression of $smad7$ mRNA leads to a decrease in the number of $isl1$-promoter-active SHF cells in the venous pole. (G) Confocal 3D-stack images (at 26 hpf) of the $TgBAC(isl1:GFP);Tg(myl7:Nls-mCherry)$ embryos treated with DMSO (upper panels) or DMH1 (10 μM, lower panels) constructed between 14 hpf and 26 hpf. Square brackets indicate the $isl1$-promoter-active SHF cells in the venous pole. (H) Quantitative analyses of the number of $isl1$-promoter-active SHF cells and of the number of both GFP-negative and $myl7$-promoter-active mCherry-positive cells. Note that DMH1 treatment decreases the number of $isl1$-promoter-active SHF cells in the venous pole but does not affect the number of GFP-negative and $myl7$-promoter-active CMs. All images in this figure are in dorsal view, anterior to the top. The confocal 3D-stack images and ISH images are a set of representative images of at least three independent experiments. **p < 0.01.

DOI: https://doi.org/10.7554/eLife.29106.020

The following source data and figure supplements are available for figure 6:

**Source data 1.** The numbers of BRE-positive cardiomyocytes in $lats1/2$ mutants at 24 hpf (*Figure 6D*), $isl1$-promoter-active SHF cells with $smad7$ mRNA injection (*Figure 6F*) and $isl1$-promoter-active SHF cells and GFP-negative CMs with DMH1 treatment (*Figure 6H*).
DOI: https://doi.org/10.7554/eLife.29106.024

**Figure supplement 1.** Depletion of Lats1/2 leads to an increase in $bmps$ expression.
DOI: https://doi.org/10.7554/eLife.29106.021

**Figure supplement 2.** Depletion of Lats1/2 leads to an activation of Bmp-Smad signaling.
DOI: https://doi.org/10.7554/eLife.29106.022

**Figure supplement 3.** Suppression of Bmp-Smad signaling results in a decrease in the number of Isl1-positive SHF cells in the venous pole.
DOI: https://doi.org/10.7554/eLife.29106.023

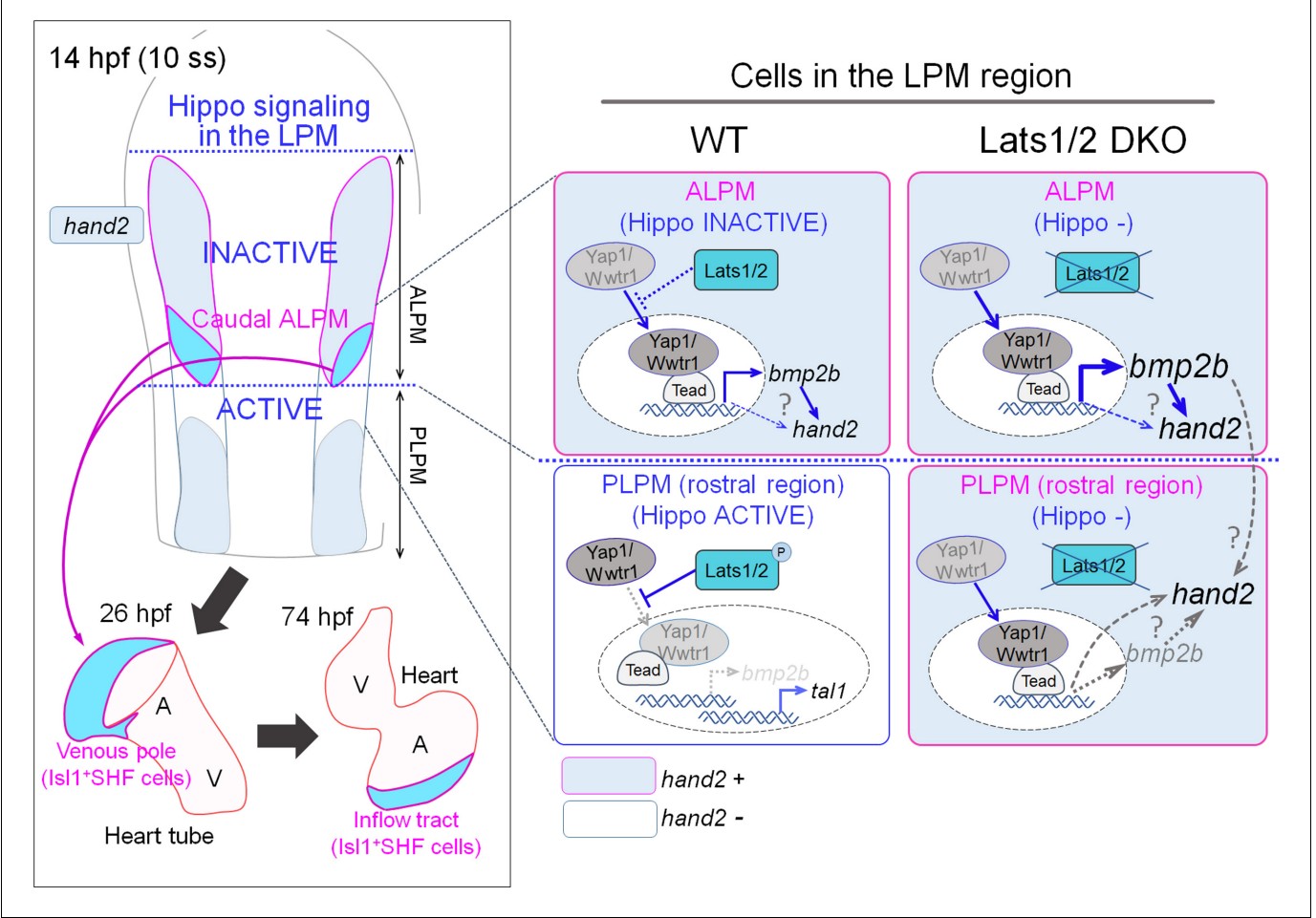

**Figure 7.** A schematic illustration of Hippo signaling in the ALPM and the border of the ALPM and PLPM in wildtype (WT) and Lats1/2 double knockout (DKO) embryos. In WT embryos, at the caudal part of the ALPM, Hippo signaling is inactive, whereas Tead-dependent transcription co-activated by Yap1/Wwtr1 is active and promotes *bmp2b* expression and *hand2* expression. Cells expressing *hand2* become the Isl1-positive SHF cells in the venous pole of the heart tube, and eventually populate the inflow tract. In the rostral region of the PLPM, Hippo signaling is active and *hand2* expression is suppressed. Cells from this region do not become cells of the venous pole. In the Lats1/2 DKO, Hippo signaling is absent in the caudal part of the ALPM, and *hand2* expression is increased. In the rostral region of the PLPM, Hippo signaling is absent and *hand2* is expressed. *hand2* expression in these cells promotes SHF specification, and these cells are integrated into the venous pole of the heart tube, and eventually populate the atrium of the heart.

DOI: https://doi.org/10.7554/eLife.29106.025

Isl1-positive, *hand2*-promoter-active CMs that are localized to the venous pole of the atrium. Both sides of the HF located caudally contributed to the venous pole of the cardiac tube. Further, mammalian SHF cells have multi-potential to differentiate into endocardial cells and smooth muscle cells in addition to myocardial cells (*Chen et al., 2009b*). By generating BAC transgenic fish, we found that *isl1*-promoter-active cells were detected in the atrial myocardium, endocardium, and epicardium, but not in the ventricular myocardium. Together with previous reports, our results suggest that the properties of zebrafish Isl1-positive SHF cells are similar to those of mammalian posterior-SHF cells.

Lats1/2-Yap1/Wwtr1-Tead signaling regulates Bmp2b expression, which is necessary for the formation of *hand2*-promoter-active cells in the ALPM and Isl1-expressing cells. Although previous reports have shown that *isl1*-positive and *mef2*-positive cells reside at the venous pole (*de Pater et al., 2009*; *Hinits et al., 2012*), the molecular mechanism explaining how ALPM cells give rise to these CPCs at the venous pole has remained unclear. To date, a number of signaling molecules, such as TGFβ, FGF, and BMP, have been reported to regulate arterial pole formation in the zebrafish

heart tube (*de Pater et al., 2009*; *Hami et al., 2011*; *Zhou et al., 2011*). We found that the Tead activation signal overlaps with the Bmp-reporter-positive signal in the ALPM. By analyzing *lats1/2* mutants and *yap1/wwtr1* mutants, we show that Hippo signaling controls *bmp2b* expression in the ALPM. Bmp-Smad inhibition expands the *tal1* expression domain to restrict LPM fate (*Gupta et al., 2006*). Interestingly, *hand2* expression is diminished in the *alk3* mutant, which is affected in a Bmp type I receptor 1a, at 12 ss (*de Pater et al., 2012*). In our experiments, Bmp-Smad inhibition results in the suppression of *hand2*-promoter-activated GFP expression at 15 hpf. Therefore, we conclude that *bmp2b* expression is positively regulated by Yap1/Wwtr1, which balances the cell fate between the HF and the blood cell field at the boundary between the ALPM and the PLPM.

We believe that the downstream targets of the Hippo-dependent Bmp-mediated signal, especially the transcription-factor-(Smads)-dependent signal, might promote distinct functions that depend on their time of action. It has been shown previously that Bmp signaling is upstream of both *hand2* and *nkx2.5* in zebrafish, and that the expression of both mRNAs is lost in Bmp signaling mutants (*de Pater et al., 2012*; *Kishimoto et al., 1997*; *Reiter et al., 2001*). However, we found here that *yap1/wwtr1* double mutants still have *hand2* expression (although this is reduced) and normal *nkx2.5* expression, even though Bmp signaling is decreased. Consistent with this possibility, Bmp signal-dependent dorso-ventral axis formation is not dependent on Lats1/2-Yap1/Wwtr1-Bmp signaling, suggesting that additional regulators of Bmp signaling are active in the ALPM. Another possibility is that Hippo signaling acts independently on both *bmp2b* expression and *hand2* expression.

In summary, we demonstrate that the Yap1/Wwtr1-Tead signal promotes Bmp2b expression and that Smad-dependent signaling subsequently defines the ALPM (*Figure 7*). Because Hippo signaling restricts the border between the ALPM and the PLPM, Hippo signaling may account for the restriction of SHF formation and the subsequent SHF-derived Isl1-positive IFT atrial CMs.

## Materials and methods

**Key resources table**

| Reagent type (species) or resource | Designation | Source or reference | Identifiers | Additional information |
|---|---|---|---|---|
| Gene (*Danio rerio*) | *yap1* | NA | ZDB-GENE-030131–9710 | PMID:25313964 |
| Gene (*Danio rerio*) | *wwtr1* | NA | ZDB-GENE-051101–1 | PMID:28350986 |
| Gene (*Danio rerio*) | *lats1* | NA | ZDB-GENE-050523–2 | PMID:19842174 |
| Gene (*Danio rerio*) | *lats2* | NA | ZDB-GENE-050119–6 | PMID:19842174 |
| Gene (*Danio rerio*) | *smad7* | NA | ZDB-GENE-030128–3 | |
| Gene (*Danio rerio*) | *bmp2b* | NA | ZDB-GENE-980526–474 | PMID:19000838 |
| Gene (*Danio rerio*) | *bmp4* | NA | ZDB-GENE-980528–2059 | |
| Gene (*Danio rerio*) | *gata4* | NA | ZDB-GENE-980526–476 | |
| Gene (*Danio rerio*) | *nkx2.5* | NA | ZDB-GENE-980526–321 | PMID:23444361 |
| Gene (*Danio rerio*) | *etv2* | NA | ZDB-GENE-050622–14 | |
| Gene (*Danio rerio*) | *tal1* | NA | ZDB-GENE-980526–501 | |
| Gene (*Danio rerio*) | *isl1* | NA | ZDB-GENE-980526–112 | PMID:19395641 |
| Gene (*Danio rerio*) | *hoxb5b* | NA | ZDB-GENE-000823–6 | PMID:19081079 |
| Strain, strain background (*Danio rerio*) | AB | ZIRC | ZDB-GENO-960809–7; RRID:ZIRC_ZL1 | |
| Genetic reagent (*Danio rerio*) | *hand2* BAC:GFP | PMID:20627079 | ZDB-ALT-110128–40; RRID:ZFIN_ZDB-ALT-110128-40 | |
| Genetic reagent (*Danio rerio*) | *hand2* BAC:tdEosFP | This paper | BacPac Resources:CH211-95C16 BAC | |
| Genetic reagent (*Danio rerio*) | *isl1* BAC:GFP | This paper | BacPac Resources:CH211-219F7 BAC | |

*Continued on next page*

*Continued*

| Reagent type (species) or resource | Designation | Source or reference | Identifiers | Additional information |
|---|---|---|---|---|
| Genetic reagent (*Danio rerio*) | *eef1a1l1*:galdb-hTEAD2ΔN-2A-mCherry | PMID:25313964 | ZDB-FISH-150901–27167 | |
| Genetic reagent (*Danio rerio*) | *myl7*:galdb-hTEAD2ΔN-2A-mCherry | This paper | | |
| Genetic reagent (*Danio rerio*) | *uas*:GFP | PMID:18202183 | | |
| Genetic reagent (*Danio rerio*) | *uas*:mRFP1 | PMID:18202183 | | |
| Genetic reagent (*Danio rerio*) | *myl7*:Nls-mCherry | PMID:25313964 | ZDB-GENO-150218–2 | |
| Genetic reagent (*Danio rerio*) | *myh6*;Nls-tdEosFP | This paper | | |
| Genetic reagent (*Danio rerio*) | *myl7*:EGFP | This paper | | |
| Genetic reagent (*Danio rerio*) | *BRE*:GFP | PMID:21337469 | ZDB-ALT-110308–1; RRID:ZFIN_ZDB-ALT-110308-1 | |
| Genetic reagent (*Danio rerio*) | *tol2 transposase* mRNA | PMID:15239961 | | 50 pg injection |
| Genetic reagent (*Danio rerio*) | *ytip-mCherry* mRNA | PMID:25313964 | | 200 pg injection |
| Genetic reagent (*Danio rerio*) | *lats1*-atg MO | PMID:19842174, GeneTools | ZDB-MRPHLNO-100415–2 | 1.2 ng injection |
| Genetic reagent (*Danio rerio*) | *lats2*-atg MO | PMID:19842174, GeneTools | ZDB-MRPHLNO-100415–4 | 1.2 ng injection |
| Genetic reagent (*Danio rerio*) | *ajuba*-atg MO | PMID:22771034, GeneTools | ZDB-MRPHLNO-120821–5 | 8 ng injection |
| Genetic reagent (*Danio rerio*) | *control* MO | Standard control oligo, GeneTools | | 5 ng injection |
| Antibody | Anti-GFP antibody | Abcam | Abcam:ab13970; RRID:AB_300798 | (1:300) |
| Antibody | Anti-mCherry antibody | Clontech | Clontech:632543; RRID:AB_2307319 | (1:300) |
| Antibody | Anti-Isl1 antibody | Genetex | Genetex:GTX128201 | (1:100) |
| Antibody | Anti-pSmad1/5/9 antibody | Cell Signaling Technology | Cell Signaling Technology:13820S | (1:100) |
| Antibody | Anti-chicken Alexa Fluor 488 IgG | Thermo Fisher Scientific | Thermo Fisher Scientific: A-11039; RRID:AB_142924 | (1:300) |
| Antibody | Anti-mouse Alexa Fluor 546 IgG | Thermo Fisher Scientific | Thermo Fisher Scientific: A-11030; RRID:AB_2534089 | (1:300) |
| Antibody | Anti-rabbit Alexa Fluor 633 IgG | Thermo Fisher Scientific | Thermo Fisher Scientific: A-21070; RRID:AB_2535731 | (1:300) |
| Recombinant DNA reagent | pCR4-bluntTOPO | Thermo Fisher Scientific | Thermo Fisher Scientific: K287520 | |
| Recombinant DNA reagent | pTol2 | PMID:18202183 | | |
| Recombinant DNA reagent | pcDNA3.1 | Thermo Fisher Scientific | Thermo Fisher Scientific: V790-20 | |
| Recombinant DNA reagent | pRedET | Gene Bridges | | |
| Recombinant DNA reagent | pCS2 | Clontech | RRID:SCR_007237 | |
| Commercial assay or kit | mMessage mMACHINE kit | Thermo Fisher Scientific | Thermo Fisher Scientific: AM1340 | |

*Continued on next page*

*Continued*

| Reagent type (species) or resource | Designation | Source or reference | Identifiers | Additional information |
|---|---|---|---|---|
| Commercial assay or kit | KOD FX Neo DNA polymerase | TOYOBO | TOYOBO:KFX-201 | |
| Commercial assay or kit | Click-iT EdU Alexa 647 Imaging Kit | Thermo Fisher Scientific | Thermo Fisher Scientific: C10340 | |
| Commercial assay or kit | DIG RNA labeling lit | Roche | Sigma-Aldrich:11175025910 | |
| Commercial assay or kit | BM-purple | Roche | Sigma-Aldrich:11442074001 | |
| Commercial assay or kit | QuantiFast SYBR Green PCR kit | Qiagen | Qiagen:204054 | |
| Chemical compound, drug | DMH1 | Calbiochem | Sigma-Aldrich:203646 | 10 µM addition |
| Chemical compound, drug | PTU | Sigma-Aldrich | Sigma-Aldrich:P7629 | |
| Software, algorithm | GraphPad Prism 7 | GraphPad Software | RRID:SCR_002798 | |
| Software, algorithm | Imaris ver.8.4.1 | Bitplane | RRID:SCR_007370 | |
| Other | MultiNA microchip electrophoresis system | SHIMADZU | | |

## Zebrafish (*danio rerio*) strains, transgenic lines, and mutant lines

The experiments using zebrafish were approved by the institutional animal committee of National Cerebral and Cardiovascular Center (permit number: 17003) and performed according to the guidelines of the Institute. We used the AB strain as wildtype. The following zebrafish transgenic lines were used for experiments: *Tg(eef1a1l1:galdb-hTEAD2ΔN-2A-mCherry)* fish (*Fukui et al., 2014*), *Tg (myl7:Nls-mCherry)* fish (*Fukui et al., 2014*), *TgBAC(hand2:GFP)* fish (*Yin et al., 2010*), *Tg(BRE:GFP)* fish (*Collery and Link, 2011*), *Tg(uas:mRFP1)* fish (*Asakawa et al., 2008*), and *Tg(uas:GFP)* fish (*Asakawa et al., 2008*). The *Tg(myl7:galdb-hTEAD2ΔN-2A-mCherry)* fish, *Tg(myh6:Nls-tdEosFP)* fish, *Tg (myl7:EGFP)* fish, and *TgBAC(isl1:GFP)* fish were generated as described in the experimental procedures. The knockout alleles *ncv107* for *lats1*, *ncv108* for *lats2*, and *ncv117* for *yap1* genes were generated by TALEN techniques as described in the experimental procedures. The *ncv114* allele for *wwtr1* was previously described by *Nakajima et al (2017)*.

## Image acquisition and image processing

To obtain the images of embryos, the pigmentation of the embryos was suppressed by the addition of 1-phenyl-2-thiourea (PTU) (Sigma-Aldrich, St. Louis, MO) into breeding E3 media. Embryos were dechorionated and mounted in 1% low-melting agarose dissolved in E3 medium. Confocal images of 2.0 µm steps were taken with a FV1200 confocal microscope system (Olympus, Tokyo, Japan) equipped with a water immersion 20x lens (XLUMPlanFL, 1.0 NA, Olympus). Images were processed with a FV10-ASW 4.2 viewer (Olympus). The distance between the *hand2*-positive region of the ALPM and the PLPM was measured using DP2-BSW software (Olympus). Cell-tracking data containing nuclei positions were analyzed using Imaris8.4.1 software (Bitplane, Zurich, Switzerland).

## Generation of knockout zebrafish by TALEN

To develop knockout zebrafish, we used transcription activator-like effector nuclease (TALEN) Targeter 2.0 (https://tale-nt.cac.cornell.edu) to design a TALEN pair that targets *lats1*, *lats2*, and *yap1*. The target sequence of TAL-*lats1*, TAL-*lats2*, and TAL-*yap1* were 5′-TCAGCAAATGCTGCAGGAGA TccgagagagcctgcgaAACCTCTCCCCGTCCTCCAA-3′, 5′-TCTCGAGGAGAGGGTGgtcgaggtggagact-CAAAGGGCAAAGACCA-3′, and 5′-CCGAACCAGCACAACCctccagccggccaccagaTCGTCCATG TTCGGGG-3′, respectively (capital letters are the sequences of the left [TAL-*lats1*-F, *lats2*-F, and *yap1*-F] and right [TAL-*lats1*-R, *lats2*-R, and *yap1*-R] arms, respectively). These expression plasmids of the TALEN-pair were constructed by pT3TS-GoldyTALEN. TALEN mRNAs were synthesized in vitro using a T3 mMessage mMACHINE kit (Thermo Fisher Scientific, Waltham, MA). To induce double-

strand breaks in the target sequence, 50 pg of TAL-*lats1-F* / -*lats1-R* mRNAs, TAL-*lats2-F* / -*lats2-R* mRNAs, or TAL-*yap1-F* / -*yap1-R* mRNAs were injected into one- to two-cell stage transgenic embryos. Each injected founder (F0) fish was outcrossed to wildtype fish to obtain the $F_1$ progeny from individual founders. The generation of *wwtr1* knockout zebrafish has been reported previously (*Nakajima et al., 2017*). To analyse TALEN-induced mutations, genomic DNA from $F_1$ embryos was lysed by 50 µl of NaOH solution (50 mM) at 95°C for 5 min, and 5 µl of Tris-HCl (pH8.0, 1.0 M) was added on ice for 10 min. After centrifugation (13,500 rpm, 5 min), PCR reaction was performed using KOD FX Neo DNA polymerase (TOYOBO, Osaka, Japan). The genotyping PCR primers were used for amplification: *lats1* (5′-GGCACTTAACATATGCTTTTACATG-3′ and 5′-TTTGCTGCTGTC TGCGGAGCTGTT-3′); *lats2* (5′-AGAGTTTGTGTGAGAGAAAACAGG-3′ and 5′-GCATTGACCAGA TCCTGTAGCATC-3′); *yap1* (5′-TCCTTCGCAAGGCTTGGATAATTG-3′ and 5′-TTGTCTGGAG TGGGACTTTGGCTC-3′); *wwtr1* (5′-GGACGAAAAACAGGAAAAGTTC-3′ and 5′-ACTGCGGCATA TCCTTGTTC-3′). These amplified PCR products were analyzed using a MCE-202 MultiNA microchip electrophoresis system (SHIMADZU, Kyoto, Japan) with the DNA-500 reagent kit (SHIMADZU).

## Microinjection of oligonucleotide and mRNA

We injected 200 pg *ytip-mCherry* mRNA (*Fukui et al., 2014*), 100 or 200 pg zebrafish-*smad7* mRNA, 1.2 ng *lats1-atg* MO (5′-CCTCGGGTTTCTCGGCCCTCCTCAT-3′) (*Chen et al., 2009a*), 1.2 ng *lats2-atg* MO (5′-CATGAGTGAACTTGGCCTGTTTTCT-3′) (*Chen et al., 2009a*), 8 ng *ajuba-atg* MO (5′-TGAGTTTGATGCCAAGTCGATCCAT-3′) (*Witzel et al., 2012*), and 5 ng *control* MO (5′-CC TCTTACCTCAGTTACAATTTATA-3′) as previously reported (*Fukui et al., 2014*). These morpholinos were purchased from Gene Tools (Philomath, OR). Capped mRNAs were synthesized using the SP6 mMessage mMachine system (Thermo Fisher Scientific). Microinjection was performed using Femto-Jet (Eppendorf, Hamburg, Germany). MOs, mRNA, and Tol2 plasmids were injected into blastomeres at the one- to two-cell stage.

## EdU incorporation assay

The *TgBAC(isl1:GFP);Tg(myl7:Nls-mCherry)* embryos injected with control MO or *lats1/2* MOs were incubated with 2 mM of 5-ethynyl-2-deoxyuridine (EdU) from 14 to 26 hpf or from 20 to 36 hpf, and subsequently fixed using 4% PFA at 96 hpf. EdU-incorporated cells were labelled by Click-iT EdU Alexa Fluor 647 Imaging Kits (Thermo Fisher Scientific) following the manufacturer's instructions. Images were taken using the FV1200 confocal microscope system. The number of EdU-positive *isl1*-promoter-active CMs was determined by counting the number of cells with overlapping Alexa Fluor 647-positive signal, *isl1*-promoter-activated signal and *myl7*-promoter-activated signal.

## Whole-mount in situ hybridization (WISH)

The antisense *hand2, isl1, bmp2b, bmp4, gata4, nkx2.5, etv2, tal1,* and *hoxb5b* RNA probes labeled with digoxigenin (DIG) were prepared using an RNA labeling kit (Roche, Basel, Switzerland). WISH was performed as previously described (*Fukui et al., 2014*). Colorimetric reaction was carried out using BM purple (Roche) as the substrate. To stop the reaction, embryos were washed with PBS-T, fixed with 4% PFA for 20 min at room temperature, and subsequently immersed in glycerol. Images were taken using a SZX-16 Stereo Microscope (Olympus).

## Immunohistochemistry

Embryos at 14 hpf and 26 hpf were fixed by MEMFA (3.7% formaldehyde, 0.1 M MOPS, 2 mM EGTA, 1 mM $MgSO_4$) for 2 hr at room temperature. After fixation, the solution was changed to 50% methanol/MEMFA for 10 min, then changed to 100% methanol at room temperature, and then stored in 100% methanol at –30°C overnight. After rehydration, embryos were washed three times for 10 min in PBBT (PBS with 2 mg/mL BSA and 0.1% TritonX-100). Embryos were blocked in PBBT with 10% goat serum for 60 min at room temperature, and subsequently incubated overnight at 4°C with primary antibodies, 1:300 diluted chicken anti-GFP antibody (ab13970, Abcam, Cambridge, UK), 1:300 diluted mouse anti-mCherry antibody (632543, Clontech, Mountain View, CA), and 1:100 diluted rabbit anti-Islet1 antibody (GTX128201, Genetex, Irvine, CA) or 1:100 diluted rabbit anti-pSmad1/5/9 antibody (13820S, Cell Signaling TECHNOLOGY, Danvers, MA) in blocking solution. Embryos were washed with PBBT five times over the course of 2 hr, with blocking solution for 60

min at room temperature, and incubated overnight at 4°C with secondary antibodies, anti-chicken Alexa Fluor 488 IgG (A-11039, Thermo Fisher Scientific), anti-mouse Alexa Fluor 546 IgG (A-11030, Thermo Fisher Scientific), and anti-rabbit Alexa Fluor 633 IgG (A-21070, Thermo Fisher Scientific) diluted 1:300 in blocking solution. Embryos were washed with PBBT five times over the course of 2 hr and stored in PBS at 4°C prior to imaging.

## Quantitative real-time PCR (q-PCR)

Total RNAs were collected from whole-embryonic cells using TRizol (Thermo Fisher Scientific) following the manufacturer's instructions. For q-PCR, reverse transcription and RT-PCR were performed with the QuantiFast SYBR Green PCR kit (Qiagen, Hilden, Germany) in the Mastercycler Realplex (Eppendorf). The following primer set was used for amplification: nkx2.5-S (5′-GCTTTTACGCGAA-GAACTTCC-3′), nkx2.5-AS (5′-GATCTTCACCTGTGTGGAGG-3′); gata4-S (5′-AAGGTCATCCCGG TAAGCTC-3′), gata4-AS (5′-TGTCACGTACACCGGAGAAG-3′); hand2-S (5′-TACCATGGCACCTTCG TACA-3′), hand2-AS (5′-CCTTTCTTCTTTGGCGTCTG-3′); eef1a1l1-S (5′-CTGGAGGCCAGC TCAAACAT-3′), eef1a1l1-AS (5′-ATCAAGAAGAGTAGTACCGCTAGCATTAC-3′) (*Fukui et al., 2014*).

Plasmids cDNA fragments encoding zebrafish Hand2, Isl1, Bmp2b, Bmp4, Gata4, Nkx2.5, Etv2, Tal1, Hoxb5b and Smad7 were amplified by PCR using a cDNA library derived from zebrafish embryos and subcloned into a pCR4 Blunt TOPO vector (Thermo Fisher Scientific). The following primer set were used for amplification: hand2-S (5′-CGGGATCCCGCCATGAGTTTAGTTGGAGGG TT-3′ [containing BamHI sequence]), hand2-AS (5′-GCTTTAGTCTCATTGCTTCAGTTCC-3′); isl1-S (5′-GCTCTAGACCTTACTTTCTTGACATGGGAGAC-3′ [containing XbaI sequence]), isl1-AS (5′-GGAC TGGTCGCCACCATTGGAGTA-3′); bmp2b-S (5′-ATGTCGACACCATGGTCGCCGTGGTCCGCGCTC TC-3′ [containing SalI sequence]), bmp2b-AS (5′-TCATCGGCACCCACAGCCCTCCACC-3′); bmp4-S (5′-CGGGATCCCATGATTCCTGGTAATCGAATGC-3′ [containing BamHI sequence]), bmp4-AS (5′-CATTTGTACAACCTCCACAGCAAG-3′); gata4-S (5′-GTGAATTCATGTATCAAGGTGTAACGA TGGCC-3′ [containing EcoRI sequence]), gata4-AS (5′-GAGCTTCATGTAGAGTCCACATGC-3′); nkx2.5-S (5′-GCTCTAGATTCCATGGCAATGTTCTCTAGCCAA-3′ [containing XbaI sequence]), nkx2.5-AS (5′-GATGAATGCTGTCGGTAAATGTAG-3′); etv2-S (5′-GTGAATTCCTGGATTTTACACA-GAAGACTTCAGA-3′ [containing EcoRI sequence]), etv2-AS (5′-CCACGACTGAGCTTCTCATAGTTC-3′); tal1-S (5′-GTGAATTCGAAATCCGAGCAATTTCCGCTGAG-3′ [containing EcoRI sequence]), tal1-AS (5′-CTTAGCATCTCCTGAAGGAGGTCGT-3′); hoxb5b-S (5′-GTGAATTCCCAAATGAGCTCTTA TTTTCTAAACTCG-3′ [containing EcoRI sequence]), hoxb5b-AS (5′-GATGTGATTTGATCAA TTTTGAAACGCGC-3′); smad7-S (5′-AGGGATCCTCCCGCATGTTCAGGACCAAACGAT-3′ [containing BamHI sequence]), smad7-AS (5′-GAAGGCCTTTATCGGTTATTAAATATGACCTCTAACC-3′ [containing StuI sequence]). The cDNA of zYtip was previously amplified and cloned into the pCS2 vector (Clontech) (*Fukui et al., 2014*). The DNA encoding Smad7 was subcloned into the pCS2 vector to construct the pCS2-smad7. All of the cDNAs amplified by PCR using cDNA libraries were sequenced. Mutations were also confirmed by sequencing.

## Generation of transgenic lines

To monitor atrial CM development, we established a transgenic (Tg) zebrafish line expressing a nuclear localization signal (Nls)-tagged tandem Eos fluorescent protein under the control of the *myosin heavy chain 6* (*myh6*) promoter; Tg(*myh6*:Nls-tdEosFP). pTol2-*myh6* vector was constructed by modifying the pTol2 vector and inserting the *myh6* promoter as a driver of the expression of the target molecule. The primers used to amplify the *myh6* promoter were 5′-AGAGCTAAAGTGGCAGTG TGCCGAT-3′ and 5′-TCCCGAACTCTGCCATTAAAGCATCAC-3′. An oligonucleotide encoding Nls derived from SV40 (PKKKRKV) was inserted into pcDNA-tdEosFP (MoBiTec, Göttingen, Germany) to generate the plasmids expressing Nls-tagged tandem Eos fluorescent protein (Nls-tdEosFP). The Nls-tdEosFP cDNA was subcloned into the pTol2-*myh6* vector to construct the pTol2-*myh6*:Nls-tdEosFP plasmids.

To monitor CM-specific Yap1/Wwtr1-dependent transcriptional activation, we developed a transgenic fish that expresses human (h) TEAD2 lacking the amino-terminus (1–113 aa) and fused with the Gal4 DNA-binding domain followed by 2A mCherry under the control of the *myosin light polypeptide 7* (*myl7*) promoter; Tg(*myl7*:galdb-hTEAD2ΔN-2A-mCherry). This Tg fish was crossed with *Tg*

(uas:GFP) reporter fish to obtain Tg(myl7:galdb-hTEAD2ΔN-2A-mCherry);Tg(uas:GFP). The pTol2-myl7 vector and the pcDNA3.1 vector containing human TEAD2ΔN cDNA fused to the DNA-binding domain of Gal4 (pcDNA3.1-galdb-hTEAD2ΔN) were constructed as previously described (Fukui et al., 2014). The Gal4db-hTEAD2ΔN cDNA was subcloned into the pTol2-myl7 vector to construct the pTol2-myl7:galdb-hTEAD2ΔN plasmids.

To monitor CM development, we developed a transgenic line that expresses EGFP under the control of the myl7 promoter; Tg(myl7:EGFP). The EGFP was subcloned into a pTol2-myl7 vector to construct the pTol2-myl7:EGFP plasmids. All of the cDNAs amplified by PCR using cDNA libraries were confirmed by DNA sequencing.

To monitor SHF development, we established a transgenic line that expressed GFP under the control of isl1 BAC promoter/enhancer; the TgBAC(isl1:GFP). pRedET plasmid (Gene Bridges, Heidelberg, Germany) was introduced into E. coli containing a CH211-219F7 BAC clone encoding the isl1 gene (BacPAC resources) by electroporation (1800V, 25 mF, 200 Ω) to increase the efficiency of homologous recombination, as previously described (Ando et al., 2016). Tol2 long terminal repeats in opposite directions flanking an ampicillin resistance cassette were amplified by PCR using Tol2_amp as a template, and these sequences were inserted into the BAC vector backbone. The cDNA encoding both GFP and a kanamycin resistance cassette (GFP_KanR) was amplified by PCR using a pCS2-GFP_KanR plasmid as a template, and inserted into the start ATG of the isl1 gene. Primers to amplify the GFP_KanR for isl1 gene were 5′-gggccttctgtccggttttaaaagtggacctaacaccgccttacttttct-tACCATGGTGAGCAAGGGCGAGGAG-3′ and 5′-aaataaacaataaagcttaacttacttttcggtggatcccc-catgtctccTCAGAAGAACTCGTCAAGAAGGCG-3′ (small letters are the homology arm to the BAC vector, whereas capital letters are the primer binding site to the template plasmid).

Tol2-mediated zebrafish transgenesis was performed by injecting 30 pg of the transgene plasmid together with 50 pg tol2 transposase mRNA, followed by subsequent screening of $F_1$ founders and establishment of single-insertion transgenic strains through selection in $F_3$ generations.

## Photoconversion

We performed photoconversion experiments by examining the transient hand2-promoter-dependent expression of tdEosFP. Tg(myl7:EGFP) embryos were injected with 30 pg of pTol2-hand2 BAC:tdEosFP plasmid along with 50 pg tol2 transposase mRNA. To trace ALPM cells, the caudal region of the left or right ALPM was photoconverted by a 405 nm laser at 15 hpf (12 ss). The photoconverted cells expressing red fluorescence were traced in the heart region at 52 hpf.

To construct the pTol2-hand2 BAC:tdEosFP, pRedET plasmid was introduced into E. coli containing a CH211-95C16 BAC clone encoding the hand2 gene (BacPAC resources) by electroporation. Tol2 long terminal repeats in opposite directions flanking an ampicillin resistance cassette were amplified by PCR using Tol2_amp as a template, and were inserted into the BAC vector backbone. The cDNA encoding tdEosFP together with a kanamycin resistance cassette (tdEosFP_KanR) was amplified by PCR using the pCS2-tdEosFP_KanR plasmid as a template, and then inserted into the start ATG of the hand2 gene. Primers to amplify the tdEosFP_KanR for the hand2 gene were 5′-ccaaagcgtactccgtctgtggttcgccgtagggtatagacaagtctgtcACCATGAAGATCAACCTCCGTATGGAAG-3′ and 5′-tagccgtcatggtgcatcacagggtggtggggaaaccctccaactaaactTCAGAAGAACTCGTCAA-GAAGGCG-3′ (small letters represent the homology arm to BAC vector, and capital letters the primer binding site to the template plasmid).

## Chemical treatment

The TgBAC(hand2:GFP);Tg(myl7:Nls-mCherry) embryos and the TgBAC(isl1:GFP);Tg(myl7:Nls-mCherry) embryos were treated with 10 μM DMH1 (203646, Sigma-Aldrich), an inhibitor of Bmp signaling, from 14 hpf to 26 hpf. As a control, embryos were incubated in E3 solution containing DMSO. These embryos were imaged at 26 hpf.

## Data analysis and statistics

Data were analyzed using GraphPad Prism 7 (GraphPad Software, La Jolla, CA). All columns shown in histograms represent a mean ± SEM. The statistical significance of multiple groups was determined by one-way ANOVA with Bonferroni's post hoc test. The numbers of atrial and ventricular

CMs at 74 hpf were analyzed by Student's t-test. The statistical significance of two groups was determined by Student's t-test.

## Acknowledgements

We thank DY Stainier for the *TgBAC(hand2:GFP)* fish; and M Sone, T Babazono, K Hiratomi, M Ueda, and S Toyoshima for their technical assistance.

## Additional information

### Funding

| Funder | Grant reference number | Author |
|---|---|---|
| Ministry of Education, Culture, Sports, Science, and Technology | 15H01221 | Hajime Fukui |
| Takeda Medical Research Foundation | | Hajime Fukui<br>Naoki Mochizuki |
| Uehara Memorial Foundation | | Hajime Fukui |
| Cell Science Research Foundation | | Hajime Fukui |
| Japan Society for the Promotion of Science | 16H02618 | Naoki Mochizuki |
| Japan Agency for Medical Research and Development | 13414779 | Naoki Mochizuki |

The funders had no role in study design, data collection and interpretation, or the decision to submit the work for publication.

### Author contributions

Hajime Fukui, Conceptualization, Resources, Data curation, Formal analysis, Funding acquisition, Investigation, Methodology, Writing—original draft, Project administration; Takahiro Miyazaki, Hiroyuki Ishikawa, Resources, Investigation; Renee Wei-Yan Chow, Writing—review and editing; Hiroyuki Nakajima, Resources, Validation, Investigation; Julien Vermot, Supervision, Writing—review and editing; Naoki Mochizuki, Conceptualization, Supervision, Funding acquisition, Project administration, Writing—review and editing

### Author ORCIDs

Hajime Fukui [ID] https://orcid.org/0000-0002-7652-2222
Julien Vermot [ID] http://orcid.org/0000-0002-8924-732X
Naoki Mochizuki [ID] http://orcid.org/0000-0002-3938-9602

### Ethics

Animal experimentation: Animal experimentation: The experiments using zebrafish were approved by the institutional animal committee of National Cerebral and Cardiovascular Center (Permit number:17003) and performed according to the guidelines of the Institute.

### Decision letter and Author response

Decision letter https://doi.org/10.7554/eLife.29106.031
Author response https://doi.org/10.7554/eLife.29106.032

## Additional files

### Supplementary files

• Transparent reporting form

DOI: https://doi.org/10.7554/eLife.29106.026

**Data availability**

All data generated or analyzed during this study are included in the manuscript.

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
