## [Decision Letter]

Thank you for submitting your article "Hippo signaling restricts cells in the second heart field that differentiate into Islet-1-positive atrial cardiomyocytes" for consideration by *eLife*. Your article has been reviewed by three peer reviewers, one of whom is a member of our Board of Reviewing Editors, and the evaluation has been overseen by Didier Stainier as the Senior Editor. The reviewers have opted to remain anonymous.

The reviewers have discussed the reviews with one another and the Reviewing Editor has drafted this decision to help you prepare a revised submission.

Summary:

In this manuscript, Fukui et al. examine the role of the Hippo signaling pathway in regulating development of the inflow tract (IFT) in the zebrafish heart. Using a series of elegant experiments and detailed quantification, the authors demonstrate that Hippo signaling is active in the developing IFT and that enhanced Hippo signaling results in an increased number of IFT cells, as labelled by hand2 and islet1 expression, as well as increased atrial cell number at later stages. The mechanism for the increase in cell number is not through proliferation but rather through increased differentiation of cardiomyocytes. Furthermore, the authors demonstrate that Bmp signaling is increased in the context of enhanced Hippo signaling, and they propose to place Bmp signaling downstream of Hippo and upstream of hand2 and islet1. Previous work has shown that Hippo signaling in mouse embryos determines heart size by regulating cardiomyocyte proliferation. In contrast, this work suggests that Hippo can also affect heart size by regulating specification of IFT cells. Thus, this study is significant on several levels: it expands our understanding of the roles of the Hippo signaling pathway in regulating heart size, it adds to our understanding of the pathways that regulate the formation of the IFT, and it has the potential to provide broader insights into the mechanisms that control of cardiac differentiation and patterning. However, certain elements of the authors' conclusions are not fully supported by the data presented here. Several issues require further investigation and/or clarification in order to make a clear case for their model.

Essential revisions:

1) Importantly, the authors need to provide further evidence to support their conclusions regarding the differential fates of the left and right caudal ALPM. Previous cell tracking work has shown that cells from the left LPM form the dorsal side of the cardiac tube while cells from the right LPM form the ventral side of the cardiac tube (PMID: 18267096). Different from the description in the current manuscript, the older work showed that both the right and left LPM cells contributed to the venous pole. However, the dorsal side of the tube extends more anteriorly compared to the ventral side. Looking closely at the data provided by the authors (e.g. by playing the last 3 hours (equivalent of 2 secs) of their movie backward and forward rapidly), one gets the impression that the right caudal ALPM also contributes to the venous pole at the ventral side. Likely due to the tissue depth at the ventral surface, the fluorescence intensity of these cells is weaker and, therefore, difficult to track. To definitively state that the left and right sides contribute to the venous and arterial poles, respectively, the authors should provide higher resolution (and 3D) tracking of both the right and left ALPM cells. In addition, complementary lineage tracing experiments (using photoconversion or other cell labeling techniques) should be employed to follow the cells labeled in the right and left ALPM to their destination in the heart at 48 hpf.

2) The authors highlight the expression of the Tead reporter in the left caudal ALPM, but it is not clear whether or not this reporter is expressed in the right caudal ALPM. This needs to be clarified.

3) The ectopic hand2 expression and bmp2 expression in lats1/2 mutants is bilateral and not restricted to the left ALPM (Figures 5 and 6). How does this reconcile with the authors' suggested laterality of Hippo signaling (as shown in their model in Figure 7)?

4) In Figure 6, the authors show enhanced expression of bmp2b in lats1/2 mutants and loss of bmp2 expression in yap/wwtr mutants. The effects are visible in the entire ALPM and head region, while their data shown in Figure 3 suggests that Yap1 is only active in the posterior ALPM. This discrepancy is not addressed. How do the authors fit these data with their model in which Hippo signaling activity is different in the posterior versus anterior ALPM?

5) It's possible that an increase in signaling through either pathway (Hippo or Bmp) would result in increased cells at the IFT. Is there increased TEAD reporter activity following bmp2b heat-shock?

6) To reduce BMP signaling, the authors inject mRNA for smad7 (inhibitory Smad). It is not clear why the authors try to block BMP signalling during early stages of development since BMP signaling is required for proper dorsal-ventral patterning. Did the authors observe DV patterning defects in these experiments?

7) The authors imply that lats1 and lats2 have a special influence on the Isl1-positive atrial cells and do not affect the numbers of the Isl1-negative atrial cells. However, this is not clearly demonstrated by the data presented here: the total number of atrial cells is presented at one stage, and the number of Isl1-positive cells is presented at a substantially earlier stage. Does the change in the number of Isl1-positive cells account for the entirety of the difference in atrial size? Similarly, is the impact of overexpressing bmp2b and/or adding DMH1 specific to the Isl1-positive population, or do these manipulations affect the entire atrium?

8) The role of Yap1/Wwtr1 in promoting hand2 expression needs further clarification. The authors point to the reduction of hand2 expression in yap1;wwtr1 mutants, but it is unclear how specifically this phenotype relates to the regulation of hand2. Is the entire ALPM reduced in yap1;wwtr1 mutants? In addition, the relationship between the cardia bifida phenotype and the reduced hand2 expression in the yap;wwtr1 mutant should be more clearly articulated. The yap;wwtr1 mutant appears smaller than the control embryo shown. Corresponding bright field images would be helpful here.

[Editors' note: further revisions were requested prior to acceptance, as described below. The authors’ plan for revisions was approved and the authors made a formal revised submission.]

Thank you for sending your revised article entitled "The Hippo signaling suppresses the differentiation of cardiac precursor cells from lateral plate mesoderm prior to establishment of the heart field in zebrafish" for peer review at *eLife*. Your revised article has been evaluated by its original three peer reviewers, one of whom is a member of our Board of Reviewing Editors, and the evaluation is being overseen by Didier Stainier as the Senior Editor.

The reviewers have discussed their reviews with one another, and their discussion has highlighted some remaining issues (elaborated below) that still need to be addressed. Since some of these revisions would likely require you to perform new experiments, we ask that you respond to this letter within the next two to three weeks with a specific action plan and timetable for the completion of the additional work. The Reviewing Editor and reviewers will then consider your plans, assess the feasibility of an effective response to their concerns, and offer a binding recommendation (such as an official invitation to submit a second revision of your manuscript to *eLife*). Alternatively, if you would prefer to submit your manuscript elsewhere, rather than revising it again for *eLife*, please let us know.

In this revised manuscript, the authors have made multiple amendments to their original submission, including new data and new interpretations, and many of these have helped to clarify issues from the original manuscript that needed to be addressed. However, several concerns remain that require further attention. Importantly, while the current version of the manuscript certainly demonstrates an influence of lats1 and lats2 on the development of the venous pole, it stops short of clearly articulating a mechanistic connection between the Hippo pathway and the differentiation of venous pole cells. Specifically, some of the data shown appear to be inconsistent with or to contradict the authors' model, as described below.

Essential revisions:

1) From the authors' data, it is clear that there is a positive relationship between Bmp signaling and the formation of venous pole cells. It is also clear that Hippo signaling has a negative influence on bmp2b expression. However, it is not yet clear whether the caudal boundary of hand2 expression in the ALPM is set by Hippo signaling repressing bmp2b expression at that location. While it may be challenging to test a model that relates Hippo signaling to bmp2b expression directly, it would be beneficial for the authors to clarify aspects of their story that are not consistent with that model. For example, while bmp2b expression is increased in lats1;2 mutants, it does not appear to be expanded, and this seems to contradict the authors' model. Can the authors clarify this, or adjust their model accordingly?

2) The authors conclude that Yap1 activity in the ALPM induces Bmp signaling and that Bmp signaling activates hand2 and Isl1 expression, but this conclusion is too simple. Published data in zebrafish have shown that Bmp signaling is upstream of both hand2 and nkx2.5. (Both are lost in Bmp signaling mutants.) Yap1/wwtr1 double mutants still have hand2 expression (although reduced) and normal nkx2.5 expression. The authors do not address whether Bmp signaling is affected in the Yap1/wwtr1 mutants using the bre:gfp reporter line (they only show that bmp2b expression is reduced in the ALPM). Neither do they try to rescue the Yap1/wwtr1 phenotype with Bmp overexpression. The proposed model suggesting that Yap1 signaling is upstream of Bmp signaling is therefore an oversimplification and needs adjustment.

3) The phenotype shown for an embryo overexpressing smad7 mRNA does not seem to be a standard BMP-induced dorsalization phenotype, which is surprising. The appearance of the embryo in Figure 6—figure supplement 3A and the fragmented cells in Figure 6H suggest that the overexpression of smad7 may be causing cell death. Additionally, the cardiac phenotype attained after smad7 overexpression was not quantified, weakening the conclusions made from this experiment. If the authors intend to demonstrate cell-autonomous epistasis, they will likely need to use another approach. (Perhaps overexpression of a dominant-negative Bmp receptor?)

4) In the experiment using DNA injection to induce mosaic Bmp2b expression (Figure 6—figure supplement 2A), the cells expressing bmp2b-2A-mCherry look fragmented, suggesting toxicity. The tissue on the left side also looks necrotic and the Yap1 reporter is very weak, while the authors state that it is unaffected. These results are not appropriate for inclusion in the manuscript. It is unclear why the authors did not use the available hsp70:bmp2b transgenic line to generate mosaic embryos through transplantation.

5) In their revised manuscript, the authors conclude that venous pole cells originate on both sides of the ALPM, but they also state that "more cells moved from the left ALPM to the venous pole compared to the cells coming from the right ALPM". It is not clear that this semi-quantitative comparison is warranted, since it does not seem as if the authors tracked all of the cells on both sides of the ALPM. In Video 1, the authors compare five cells on the left to two cells on the right and state that this is representative of six independent experiments. How do the numbers compare among the six experiments? Is this a quantitatively reproducible observation? Do the photoconversion experiments (Figure 3D) reinforce the view that more venous pole cells come from the left than from the right? It is challenging to make a quantitative conclusion from this type of mosaic analysis. The authors should clarify the basis for this quantitative comparison between the left and right sides, or, alternatively, adjust their interpretation. Additionally, it would be helpful if the authors could clarify what criteria they used to define whether or not a tracked cell (or a photoconverted cell) became part of the venous pole in these experiments. Do the seven tracked cells in Video 1 account for the entirety of the venous pole?

6) The authors have revised their title and Abstract to accompany the changes to their model that they have incorporated into their revised manuscript. It may be beneficial to consider revising these further in order to better emphasize the key points that are most strongly supported by the current text. For example, the title claims that Hippo signaling "suppresses the differentiation of cardiac precursor cells" but it is not clear that suppression of differentiation is shown here. More venous pole cells are produced when Hippo signaling is inhibited, but it is not evident whether this occurs through alteration of differentiation (as opposed to alteration of specification or proliferation). To make this claim, the authors would need to provide additional data in support of this conclusion. Also, the authors emphasize that the activity of Hippo signaling occurs "prior to establishment of the heart field", but it is not clear what they mean by this phrase. When do they consider the heart field to be established, and what is the significance of its establishment as it relates to the conclusions of their manuscript? Overall, it would be more effective if the title and Abstract emphasized formation of the venous pole specifically, as that seems to be the focus of the manuscript.

[Editors' note: further revisions were requested prior to acceptance, as described below.]

Thank you for resubmitting your work entitled "Hippo signaling determines the number of atrial cells that originate from the anterior lateral plate mesoderm in zebrafish" for further consideration at *eLife*. Your revised article has been favorably evaluated by Didier Stainier (Senior editor), and three reviewers, one of whom is a member of our Board of Reviewing Editors.

The manuscript has been improved through its second round of revision, both by adding new data and by removing less convincing elements. Altogether, the revisions have streamlined and strengthened the overall clarity of the message regarding the impact of the Hippo signaling pathway on the number of venous pole cells in zebrafish, while toning down some of the more speculative or inconsistent aspects. However, there are some remaining issues that need to be addressed, as outlined below:

1) With each revision of this manuscript, the authors have modified their title and Abstract to accompany the changes in their message. Further modification would help the current title to fit more closely with the main point of the paper, which revolves around the pathways regulating the number of venous pole (Isl1+) cells. The new title and Abstract seem to emphasize the whole atrium, even though the take-home message of the data is focused on the venous pole. Further adjustment is needed to align the title and Abstract with the rest of the manuscript.

2) The data that are provided to argue that overexpression of smad7 (as in Figure 6) is non-toxic are not convincing. The authors have undertaken control experiments, examining whether embryos injected with smad7 mRNA look healthy, but this context is quite different from the mosaic scenario presented in the experiments, in which three different mRNAs are co-injected. It remains unexplored whether this cocktail is causing cell death (and therefore loss of hand2 expression) and therefore no conclusion can be drawn from this experiment. Therefore, the experiments utilizing smad7 overexpression should be removed from the manuscript.

3) It appears premature for the authors to conclude a direct, linear relationship connecting Hippo signaling, Bmp signaling, and Hand2 expression. Without a formal epistasis experiment (e.g. DMH1 treatment in lats1/2 LOF and measuring the hand2 response), it does not seem appropriate to suggest this hierarchy. The most appropriate representation would be to conclude that Hippo signaling acts on both Bmp signaling and Hand2 expression. These could be independent influences, or they could be linked, but the data do not clearly support the direct linkage.

4) The authors should look through the manuscript carefully, as there are a number of errors in the text. Some examples, spotted by reviewers, are listed here:

- Instances where "specification" is used, when differentiation is appropriate

- "compared to humans" would read better as "compared to mouse and humans", given it follows a description of mice

- "transcriptional factor" should read "transcription factor"

- "restricts determines" is tautological and needs altering

- "frameshifts" should read "frameshift mutations"

- "faithfully recapitulates isl1 expression in vivo" is an overstatement if all isl1 expression was not assessed.

- “Interestingly, the gap length of WT embryos was significantly shorter than that of the lats1wt/ncv107lats2ncv108 embryos, the lats1/2 DKO embryos and the lats1/2 morphants.” However, the data in Figure 5 shows that gap length in wt embryos is larger.

5) It would be beneficial for the authors to review their choices of cited articles. In some cases, it seems as if the citations may not be the best choices to fully support the statements made. For example, does Hami et al. (2011) strongly support the points for which it is cited in the Introduction (that SHF cells come from the lateral and caudal ALPM, and that Isl1+ cells give rise to the inflow tract)?

---

## [Author Response]

Essential revisions:1) Importantly, the authors need to provide further evidence to support their conclusions regarding the differential fates of the left and right caudal ALPM. Previous cell tracking work has shown that cells from the left LPM form the dorsal side of the cardiac tube while cells from the right LPM form the ventral side of the cardiac tube (PMID: 18267096). Different from the description in the current manuscript, the older work showed that both the right and left LPM cells contributed to the venous pole. However, the dorsal side of the tube extends more anteriorly compared to the ventral side. Looking closely at the data provided by the authors (e.g. by playing the last 3 hours (equivalent of 2 secs) of their movie backward and forward rapidly), one gets the impression that the right caudal ALPM also contributes to the venous pole at the ventral side. Likely due to the tissue depth at the ventral surface, the fluorescence intensity of these cells is weaker and, therefore, difficult to track. To definitively state that the left and right sides contribute to the venous and arterial poles, respectively, the authors should provide higher resolution (and 3D) tracking of both the right and left ALPM cells. In addition, complementary lineage tracing experiments (using photoconversion or other cell labeling techniques) should be employed to follow the cells labeled in the right and left ALPM to their destination in the heart at 48 hpf.

As the reviewers critically commented, the contribution of both sides of the cells in the ALPM to venous pole should be carefully examined. We have performed additional cell tracking experiments that show that the cells in the caudal regions of both sides of ALPM move to the venous pole (New Figure 3A, 3B, and Video 1). The number of the cells from the left ALPM was greater than those from the right ALPM. The cells in the both sides of LPM move toward the cardiac disk. Those cells stay in the arterial pole. Therefore, the cells in the both sides of ALPM can become both venous and arterial pole cells. In the revised manuscript, we decline our previous conclusion that the cells in the left and right ALPM become venous and arterial pole cells, respectively. Instead, we state that both sides of ALPM cells become the cells in the venous pole. Initially, we tried to understand the origin of the IFT of the atrium. Therefore, we still focus on the venous pole cells that give rise to the IFT cells.

In addition, following the reviewer’s advice, we have performed the photoconversion experiments using the embryos transiently injected with *hand2BAC*:tdEosFP and photoconverted at 12 ss. We have confirmed that both the left and right ALPM cells are localized in the inflow tract (IFT) of the atrium at 52 hpf (New Figure 3D). Therefore, we have concluded that the cells in the caudal region of the left ALPM cells contribute to venous pole and inflow tract development more than those in the right ALPM. These results are consistent with the previous report, although it is worth mentioning that more cells in the left ALPM move to the venous pole than those in the right ALPM.

2) The authors highlight the expression of the Tead reporter in the left caudal ALPM, but it is not clear whether or not this reporter is expressed in the right caudal ALPM. This needs to be clarified.

We admit that we should clarify the expression of Tead reporter in the both sides of ALPM. To examine whether the expression of Tead reporter is observed in the right side of the caudal region of the ALPM, we crossed *TgBAC(hand2:GFP)* with *Tg(eef1a1l1:galdb-hTEAD2ΔN-2A-mCherry);Tg(uas:mRFP1)*. We found that Tead reporter is expressed in both the left and right sides of entire ALPM (New Figure 3E). Collectively, our new data demonstrate that Hippo signaling functions in the entire ALPM. Therefore, this data suggests – together with the results showing the *hand2* promoter-activated cells – that Hippo signaling in the ALPM contribute to the *hand2*-positive cell formation in the ALPM.

3) The ectopic hand2 expression and bmp2 expression in lats1/2 mutants is bilateral and not restricted to the left ALPM (Figures 5 and 6). How does this reconcile with the authors' suggested laterality of Hippo signaling (as shown in their model in Figure 7)?

As the reviewers were concerned, given that the results of new experiments suggest that Hippo signaling functions in bilaterally, we have declined our previous conclusion and propose that hippo signaling functions in both the left and the right ALPM (New revised Figure 7) in the revised manuscript. This conclusion is consistent with our previous data showing the bilateral expression of both *hand2* and *bmp2b* in the ALPM as the reviewers pointed out in the *lats1/2* mutants.

4) In Figure 6, the authors show enhanced expression of bmp2b in lats1/2 mutants and loss of bmp2 expression in yap/wwtr mutants. The effects are visible in the entire ALPM and head region, while their data shown in Figure 3 suggests that Yap1 is only active in the posterior ALPM. This discrepancy is not addressed. How do the authors fit these data with their model in which Hippo signaling activity is different in the posterior versus anterior ALPM?

In the previous experiments, we just focused on the caudal region of the ALPM using the certain layer cells as the reviewers pointed out. We admit that we should carefully track the entire ALPM cells and the expression of Tead reporter as in the reply to the comments (1) and (2). In new Figure 3E and new Figure 3—figure supplement 1A, Tead expression is found in the entire ALPM but not in the rostral PLPM. Therefore, these results are consistent with the results of the enhanced expression of *bmp2b* in *lats1/2* mutant and loss of *bmp2b* expression in *yap1/wwtr1* mutants at the same developmental stage (Figure 6B), suggesting that Hippo signaling restricts the expression of *bmp2b* in the ALPM. These results were included in the model of new Figure 7.

5) It's possible that an increase in signaling through either pathway (Hippo or Bmp) would result in increased cells at the IFT. Is there increased TEAD reporter activity following bmp2b heat-shock?

Exactly, the number of the cells at the venous pole correlates with that of the IFT. Indeed, the results shown in Figure 1 prompted us to examine the cell number at the venous pole that is determined by Hippo signaling. We have demonstrated that both Tead and Bmp reporters are activated at the venous pole at 24 hpf (Figure 1—figure supplement 2B, and Figure 6E).

According to the reviewers’ suggestion, we have over-expressed *bmp2b* by the injection of *hsp70l:bmp2b-2A-mCherry* in the Tead reporter embryos. In line with the previous results, *bmp2b* expression did not affect Tead reporter activity (New Figure 6—figure supplement 2A). This new data supports our model that Hippo signaling functions upstream of Bmp-Smad signaling.

6) To reduce BMP signaling, the authors inject mRNA for smad7 (inhibitory Smad). It is not clear why the authors try to block BMP signalling during early stages of development since BMP signaling is required for proper dorsal-ventral patterning. Did the authors observe DV patterning defects in these experiments?

We admit that we did not have any better method to spatio-temporally block BMP signaling in the ALPM. Therefore, *smad7* mRNA injection was the only possible way for us to inhibit BMP signaling (Figure 6G and H). We have examined the phenotype of general inhibition of Bmp signaling, dorsalized development. As shown in the New Figure 6—figure supplement 3A and 3B, injection of *smad7* mRNA dose-dependently increased dorsalization of the embryos. Therefore, we conclude that *smad7* mRNA can block Bmp-Smad signaling to examine the Bmp-dependent signal for *hand2* expression.

7) The authors imply that lats1 and lats2 have a special influence on the Isl1-positive atrial cells and do not affect the numbers of the Isl1-negative atrial cells. However, this is not clearly demonstrated by the data presented here: the total number of atrial cells is presented at one stage, and the number of Isl1-positive cells is presented at a substantially earlier stage. Does the change in the number of Isl1-positive cells account for the entirety of the difference in atrial size? Similarly, is the impact of overexpressing bmp2b and/or adding DMH1 specific to the Isl1-positive population, or do these manipulations affect the entire atrium?

Unfortunately we cannot prove the direct relevance of the number of *isl1*-posoitve cells in the earlier stage to that of atrial cardiomyocytes in the later stage, because we have neither Tg line nor imaging system that allow us to 3-dimesionally track the cell fate from 20 hpf to 3 dpf. To show the direct contribution of Isl1-positive SHF cells to the formation of atrial cardiomyocytes, we further need to cell track of *isl1* promoter*-*activated cells. In our data, the actual increase of the atrial CM was about 20 cells in the *lats1/2* mutants at 74 hpf (Figure 1B). This increase paralleled the increase in the *isl1* promoter-activated cells at 26 hpf (Figure 4E). In addition, the number of *isl1* promoter-activated cell at 26 hpf and 96 hpf was about 25 cells (Figure 4C and 4E), suggesting that *isl1*-positive cells does not increase during this period, although we cannot exclude the possibility that *isl1* promoter-activated cells become negative, while *isl1*-negative cells could become positive. At least, these data imply that Hippo signaling determines the number of *isl1*-positive cells that give rise to atrial cardiomyocytes.

We thank the reviewers for suggesting an experiment that supports our conclusions. According to the reviewers’ advice, we examined the effect of inhibition of Bmp signaling on *isl1*-positive SHF cells and *isl1*-negative cardiomyocytes. Inhibition of Bmp signaling resulted in a decrease in the number of *isl1*-positive cells but not in that of *myl7*-positive cardiomyocytes (New Figure 6I and 6J). Therefore, we assume that the increased number of atrial cardiomyocytes in the *lats1/2* mutants is ascribed to the increased number of *isl1*-positive cells

8) The role of Yap1/Wwtr1 in promoting hand2 expression needs further clarification. The authors point to the reduction of hand2 expression in yap1;wwtr1 mutants, but it is unclear how specifically this phenotype relates to the regulation of hand2. Is the entire ALPM reduced in yap1;wwtr1 mutants? In addition, the relationship between the cardia bifida phenotype and the reduced hand2 expression in the yap;wwtr1 mutant should be more clearly articulated. The yap;wwtr1 mutant appears smaller than the control embryo shown. Corresponding bright field images would be helpful here.

Following the reviewers’ suggestion, we looked at the entire ALPM by simultaneous monitoring bright field images and confirmed that *hand2* expression is significantly reduced in the entire anterior region (New Figure 2—figure supplement 2C). In the paper by the group of Prof. Yelon (Development, 2000), *hand2* mutants (*han*) show a reduction of *hand2* expression and exhibit cardia bifida, suggesting that the reduced *hand2*-dependent signaling leads to cardia bifida. The *yap1/wwtr1* mutants showed reduced *hand2* expression and cardia bifida (Figure 5D, new Figure 2—figure supplement 2C and 2D). We further found that the expression of *nkx2.5*, one of ALPM-derived FHF marker, was not affected in *yap1/wwtr1* mutants (New Figure 2—figure supplement 2D). This data indicates that Hippo signaling is not involved in the differentiation of FHF-derived cardiomyocyte but inhibits Yap1/Wwtr1-dependent *hand2* expression in the ALPM.

[Editors' note: further revisions were requested prior to acceptance, as described below.]

Essential revisions:1) From the authors' data, it is clear that there is a positive relationship between Bmp signaling and the formation of venous pole cells. It is also clear that Hippo signaling has a negative influence on bmp2b expression. However, it is not yet clear whether the caudal boundary of hand2 expression in the ALPM is set by Hippo signaling repressing bmp2b expression at that location. While it may be challenging to test a model that relates Hippo signaling to bmp2b expression directly, it would be beneficial for the authors to clarify aspects of their story that are not consistent with that model. For example, while bmp2b expression is increased in lats1;2 mutants, it does not appear to be expanded, and this seems to contradict the authors' model. Can the authors clarify this, or adjust their model accordingly?

We agree with the reviewer that this needs clarification. *bmp2b* expression is actually very difficult to quantify and we did not obtain conclusive results when measuring its expression domain. Due to the presence of the eye near the *bmp2b* expression domain and the necessity of dissecting the yolk, the measure of the expression domain highly variable from embryo to embryo. However, we found that *bmp2b* expression levels in the *lats1/2* mutants examined by *in situ* hybridization at 10 somite-stage (ss) were greater than that of the control (Figure 6B). We further show that *bmp2b* expression was increased two-fold in *lats1/2* morphants compared to controls via qPCR performed at 24 hpf (see Author response image 1).

Bmp2b being a secreted protein, we hypothesize that increased Bmp2b expression levels might lead to an expanded BMP signaling domain as a result of an increased protein accumulation, thus leading to the activation of *hand2* expression more caudally in the ALPM. Since this hypothesis remains speculative at this point, we eliminated the previous statement “*bmp2b* expression is induced by Lats1/2-Yap1/Wwtr1 signaling in the rostral region of PLPM” in the revised manuscript.

**Author response image 1. respfig1:** Quantitative-PCR analyses of expression of bmp2b mRNAs in the whole embryos at 24 hpf injected with the lats1/2 MOs (n=4). Relative expression of mRNA in the morphants to that of the control is calculated.

We changed the model accordingly by deleting the description of cell autonomous signaling because we cannot exclude the possibility that BMP secreted from the ALPM cells might affect BMP signaling in a non-cell autonomous manner.

2) The authors conclude that Yap1 activity in the ALPM induces Bmp signaling and that Bmp signaling activates hand2 and Isl1 expression, but this conclusion is too simple. Published data in zebrafish have shown that Bmp signaling is upstream of both hand2 and nkx2.5. (Both are lost in Bmp signaling mutants.) Yap1/wwtr1 double mutants still have hand2 expression (although reduced) and normal nkx2.5 expression. The authors do not address whether Bmp signaling is affected in the Yap1/wwtr1 mutants using the bre:gfp reporter line (they only show that bmp2b expression is reduced in the ALPM). Neither do they try to rescue the Yap1/wwtr1 phenotype with Bmp overexpression. The proposed model suggesting that Yap1 signaling is upstream of Bmp signaling is therefore an oversimplification and needs adjustment.

The reviewers are right; this point deserves clarification. A possible explanation for the discrepancy of Bmp2b-dependent transcription of both *hand2* and *nkx2.5* between the previous reports and our results is that Yap1/Wwtr1-Bmp2b downstream signal varies spatially and temporally to regulate heart formation. We believe that the downstream targets of Bmp-mediated signal, especially the transcription factors (Smads)-dependent signal, promote distinct targets depending on their time of action. In support of this possibility, the Tead reporter’s activation is observed in the whole ALPM at 10 ss (New Figure 4E), whereas it was only active in the cells of the venous pole at 24 hpf (Figure 1—figure supplement 2B).

In order to address this critical point further, we performed additional experiments to define more precisely whether Yap1/Wwtr1 signaling is upstream of Bmp signaling. We followed the editor’s suggestion and studied the *Tg(BRE:GFP)* fish in the *yap1/wwtr1* heterozygous allele. Although we tried to incross the *yap1/wwtr1* heterozygous fish with the *Tg(BRE:GFP)* reporter, we could not get any viable embryos because the eggs were not fertilized. After many attempts, we concluded that our females were not fertile, possibly because of the presence of the *Tg(BRE:GFP)* transgene. Therefore, we alternatively investigated the downstream signaling of Bmp by performing immunostainings for phosphorylated Smad1/5/9 in *yap1/wwtr1* mutant, *lats1/2* mutant, and wild-type embryos (New Figure 6—figure supplement 1A). In the *yap1/wwtr1* double mutant embryos, phosphorylated Smad1/5/9 was decreased, while in the *lats1/2* double mutant embryos phosphorylated Smad1/5/9 was increased. These new data suggest that Hippo signaling in the ALPM suppresses Bmp signaling and confirm our previous conclusions.

In addition, following the reviewers’ advice, we tried to investigate whether injection of *bmp2b* mRNA rescues the *yap1/wwtr1* mutant phenotypes. As *bmp2b* overexpression leads to ventralization, which is not compatible with the analysis of the *bmp2b* expression domain, we established the optimal concentration of *bmp2b* to inject in the embryo leading to significantly low ventralized phenotype. We found that 30% of the embryos showed the ventralized phenotype when using 1.5 pg of *bmp2b* (as shown in Author response image 2). When injected in mutant embryos, we could not detect a clear rescue of the global *yap1/wwtr1* mutant morphology. Since the number of double mutants is only 1 out of 16, the number of embryos we could analyze was too low to directly assess the ALPM phenotype. Therefore, we could not draw definitive conclusions based on these experiments. This experiment requires a more complex approach such as tissue-specific expression of *bmp2b* in the ALPM using *gata4* or *nkx2.5* promoter in the double mutant. Unfortunately, providing these experiments in the time frame of the revision agreed with the editor is not possible since it requires the generation of new transgenic lines that are not available at the moment.

In light of the fact that *nkx2.5* expression is not altered in the *yap1/wwtr1* (as well as in the *lats1/2* morphants), we agree with the reviewers that our model needs adjustment. We have now changed the Discussion so that it is clear that it is possible that the expression of *nkx2.5* might be regulated by the Bmp signaling that is not downstream of the Hippo signaling. The fact that dorso-ventral axis formation is not solely dependent on the Yap1/Wwtr1-Bmp signaling supports the possibility for additional regulators of BMP signaling in the ALPM. Overall, we think our proposition that the Hippo signaling-dependent Bmp signaling acts through Hand2 expression to restrict ALPM cell fate to become venous pole cells is valid. The identity of pathways involved and their timing of action is an active line of research in the lab and will require more time to reach conclusions.

**Author response image 2. respfig2:** Bright field images of the embryo injected with bmp2b mRNA (1.5 pg) and yap1ncv117wwtr1ncv114 embryo at 22 hpf. Lateral view, anterior to the left. bmp2b mRNA leads to ventralized phenotype (left panel). Yap1/wwtr1 mutant exhibits the defect of posterior body elongation (right panel).

3) The phenotype shown for an embryo overexpressing smad7 mRNA does not seem to be a standard BMP-induced dorsalization phenotype, which is surprising. The appearance of the embryo in Figure 6–figure supplement 3A and the fragmented cells in Figure 6H suggest that the overexpression of smad7 may be causing cell death. Additionally, the cardiac phenotype attained after smad7 overexpression was not quantified, weakening the conclusions made from this experiment. If the authors intend to demonstrate cell-autonomous epistasis, they will likely need to use another approach. (Perhaps overexpression of a dominant-negative Bmp receptor?)

We sincerely apologize for the confusion. It is due to the fact that the phenotype we obtained was variable and chose to display the mild phenotype embryos to exemplify dorsalized phenotype in the embryos injected with *smad7* mRNA in the previous figure. We replaced the previous figure with a new supplemental figure (Figure 6—figure supplement 3A) showing a dorsalized embryo following the *smad7* mRNA Injection. We quantified again the percentage of dorsalized-phenotype and this time only embryos with phenotypes shown in Figure 6—figure supplement 3B were considered dorsalized embryos. The number of dorsalized embryos is decreased by comparison to the previous figure but this does not alter our conclusions that 100 pg is the best concentration to perform the overexpression experiment in the *Tg(isl1:GFP);Tg(myl7:Nls-mCherry)* embryos.

As suggested by the reviewers, we now provide a revised version where we quantified the number of *isl1*+ SHF cells in the embryo injected with *smad7* mRNA (new Figure 6F). While the number of *isl1*+ SHF cells was decreased by the injection of *smad7* mRNA, we could not detect any fragmented cells by confocal microscopy of the heart region highlighted by the *TgBAC(isl1:GFP);Tg(myl7:Nls-mCherry)* reporters at 26 hpf (see Author response image 3). These results indicate that the decrease of the *isl1*+ SHF cells of the embryo injected with *smad7* mRNA was due to the inhibition of Bmp signaling and not cell fragmentation. These results are consistent with our previous results showing the decrease of the *isl1*+ SHF cells treated with the BMP inhibitor DMH1 (Figures 6I and 6J).

Following the reviewers’ comment, we toned down our previous statement about cell- autonomous epistasis because Bmp is a secretory molecule and we cannot exclude non-cell autonomous regulation.

4) In the experiment using DNA injection to induce mosaic Bmp2b expression (Figure 6—figure supplement 2A), the cells expressing bmp2b-2A-mCherry look fragmented, suggesting toxicity. The tissue on the left side also looks necrotic and the Yap1 reporter is very weak, while the authors state that it is unaffected. These results are not appropriate for inclusion in the manuscript. It is unclear why the authors did not use the available hsp70:bmp2b transgenic line to generate mosaic embryos through transplantation.

We agree with the reviewer that the cells look fragmented. We performed new experiments by imaging the bmp2b positive cells using confocal microscopy at high magnification and these experiments confirmed that a significant fraction of the cells expressing bmp2b-2A-mCherry is fragmented. We thank the reviewers for noticing this oversight. As the reviewers suggested, the data have been deleted from the revised version. We did not proceed further with this approach, we thus decided to delete the previous Figure 6C, 6D, and Figure 6—figure supplement 2A.

**Author response image 3. respfig3:** The number of fragmented isl1 promoter-activated SHF cells of the embryos injected with the smad7 mRNA is very low as indicated at the bottom. The heart region was highlighted by the TgBAC(isl1:GFP);Tg(myl7:Nls-mCherry) embryos at 26 hpf (n=3).

Nevertheless, we strongly believe that our conclusions are still valid in absence of these experiments. Collectively, we now have the following data supporting our claim that Bmp2b is regulated by Hippo signaling to determine the number of SHF cells in the venous pole.

1) The Hippo signaling pathway inhibits phosphorylation of Smad1/5/9 (New Figure 6—figure supplement 1A).

2) The forced expression of *smad7* mRNA results in a decrease of SHF cells (New Figures 6E and 6F).

3) DMH1 treatment leads to a decrease of SHF cells (Figures 6I and 6J).

4) Both the activity of the Bmp reporter and the phosphorylation of Smad1/5/9 are increased in the venous pole (New Figures 6C, 6D, and Figure 6—figure supplement 2B).

Therefore, we feel confident that the data obtained by transient and mosaic expression of Bmp2b using Tol2 system are unnecessary to support our claim.

5) In their revised manuscript, the authors conclude that venous pole cells originate on both sides of the ALPM, but they also state that "more cells moved from the left ALPM to the venous pole compared to the cells coming from the right ALPM". It is not clear that this semi-quantitative comparison is warranted, since it does not seem as if the authors tracked all of the cells on both sides of the ALPM. In Video 1, the authors compare five cells on the left to two cells on the right and state that this is representative of six independent experiments. How do the numbers compare among the six experiments? Is this a quantitatively reproducible observation? Do the photoconversion experiments (Figure 3D) reinforce the view that more venous pole cells come from the left than from the right? It is challenging to make a quantitative conclusion from this type of mosaic analysis. The authors should clarify the basis for this quantitative comparison between the left and right sides, or, alternatively, adjust their interpretation. Additionally, it would be helpful if the authors could clarify what criteria they used to define whether or not a tracked cell (or a photoconverted cell) became part of the venous pole in these experiments. Do the seven tracked cells in Video 1 account for the entirety of the venous pole?

We agree with the reviewers’ comment that it is challenging to make quantitative conclusions about the proportion of cells coming from the left and right ALPM. We also agree that we cannot conclude about a potential bias between left and right contribution of ALPM cells into the SHF. It was a mistake on our side and we sincerely apologize for it. In fact, re-examining our data we find that the Tead reporter is bilaterally expressed equally between the two embryonic sides, which would be difficult to reconcile with our previous conclusion. Similarly, *bmp2b* expression is symmetrically altered in the *lats1/2* mutants and morphants. We thus feel that exploring the asymmetric contribution of ALPM cell to the SHF is not in the focus of our study and unnecessary. We, therefore, decided not to include this conclusion and changed the previous statement “the more cells in the left ALPM migrate toward the venous pole cells” into “*hand2* positive cells of venous pole differentiate from the caudal ALPM”.

6) The authors have revised their title and Abstract to accompany the changes to their model that they have incorporated into their revised manuscript. It may be beneficial to consider revising these further in order to better emphasize the key points that are most strongly supported by the current text. For example, the title claims that Hippo signaling "suppresses the differentiation of cardiac precursor cells" but it is not clear that suppression of differentiation is shown here. More venous pole cells are produced when Hippo signaling is inhibited, but it is not evident whether this occurs through alteration of differentiation (as opposed to alteration of specification or proliferation). To make this claim, the authors would need to provide additional data in support of this conclusion. Also, the authors emphasize that the activity of Hippo signaling occurs "prior to establishment of the heart field", but it is not clear what they mean by this phrase. When do they consider the heart field to be established, and what is the significance of its establishment as it relates to the conclusions of their manuscript? Overall, it would be more effective if the title and Abstract emphasized formation of the venous pole specifically, as that seems to be the focus of the manuscript.

We agree that the previous title, Abstract and model did not fully reflect our results as the editor pointed out. We changed the title and revised the Abstract in accordance with our results, which demonstrate the involvement of Hippo signaling in the LPM and subsequent venous pole formation. We agree that the involvement of Hippo signaling specifically in the heart field is not established in our study. Therefore, we changed the previous title into a more precise title with a focus on the venous pole specification. The new title is “Hippo signaling determines the number of atrial cells that originate from the anterior lateral plate mesoderm in zebrafish”.

We also agree that the heart field was not well introduced leading to confusion about the origin of the additional cardiomyocytes observed in the Hippo pathway mutants. We rewrote extensively the Introduction and the Results to clarify this important issue.

[Editors' note: further revisions were requested prior to acceptance, as described below.]

The manuscript has been improved through its second round of revision, both by adding new data and by removing less convincing elements. Altogether, the revisions have streamlined and strengthened the overall clarity of the message regarding the impact of the Hippo signaling pathway on the number of venous pole cells in zebrafish, while toning down some of the more speculative or inconsistent aspects. However, there are some remaining issues that need to be addressed, as outlined below:1) With each revision of this manuscript, the authors have modified their title and Abstract to accompany the changes in their message. Further modification would help the current title to fit more closely with the main point of the paper, which revolves around the pathways regulating the number of venous pole (Isl1+) cells. The new title and Abstract seem to emphasize the whole atrium, even though the take-home message of the data is focused on the venous pole. Further adjustment is needed to align the title and Abstract with the rest of the manuscript.

We thank the editors and reviewers for the helpful suggestions. We changed the title and Abstract to focus on the venous pole as they pointed out. The new title is “Hippo signaling determines the number of venous pole cells that originate from the anterior lateral plate mesoderm in zebrafish”. In the Abstract, we changed the final sentence to “Hippo signaling defines venous pole cardiomyocyte number by […]” to focus on the venous pole.

2) The data that are provided to argue that overexpression of smad7 (as in Figure 6) is non-toxic are not convincing. The authors have undertaken control experiments, examining whether embryos injected with smad7 mRNA look healthy, but this context is quite different from the mosaic scenario presented in the experiments, in which three different mRNAs are co-injected. It remains unexplored whether this cocktail is causing cell death (and therefore loss of hand2 expression) and therefore no conclusion can be drawn from this experiment. Therefore, the experiments utilizing smad7 overexpression should be removed from the manuscript.

Following the comments, to avoid the ambiguity of our results, we deleted the mosaic experiment results (Former Figure 6 G and H). Therefore, we renewed Figure 6.

3) It appears premature for the authors to conclude a direct, linear relationship connecting Hippo signaling, Bmp signaling, and Hand2 expression. Without a formal epistasis experiment (e.g. DMH1 treatment in lats1/2 LOF and measuring the hand2 response), it does not seem appropriate to suggest this hierarchy. The most appropriate representation would be to conclude that Hippo signaling acts on both Bmp signaling and Hand2 expression. These could be independent influences, or they could be linked, but the data do not clearly support the direct linkage.

We agree with the reviewer’s comment. We cannot exclude the possibility that Hippo signaling directly regulates hand2 expression and cannot conclude that hand2 is downstream of Bmp2 signaling. Therefore, we added a sentence to clarify this in the Discussion and edited Figure 7. In the new Figure 7, the two arrows denote the possibility that Hippo signaling acts on both Bmp signaling and Hand2.

4) The authors should look through the manuscript carefully, as there are a number of errors in the text. Some examples, spotted by reviewers, are listed here:- Instances where "specification" is used, when differentiation is appropriate- "compared to humans" would read better as "compared to mouse and humans", given it follows a description of mice- "transcriptional factor" should read "transcription factor"- "restricts determines" is tautological and needs altering- "frameshifts" should read "frameshift mutations"*- "faithfully recapitulates isl1 expression* in vivo" is an overstatement if all isl1 expression was not assessed.- “Interestingly, the gap length of WT embryos was significantly shorter than that of the lats1wt/ncv107lats2ncv108 embryos, the lats1/2 DKO embryos and the lats1/2 morphants.” However, the data in Figure 5 shows that gap length in wt embryos is larger.

We thank the reviewers/editors for listing up our mistakes in the manuscript. We carefully checked the manuscript including the mistakes listed up and corrected other grammatical errors.

5) It would be beneficial for the authors to review their choices of cited articles. In some cases, it seems as if the citations may not be the best choices to fully support the statements made. For example, does Hami et al. (2011) strongly support the points for which it is cited in the Introduction (that SHF cells come from the lateral and caudal ALPM, and that Isl1+ cells give rise to the inflow tract)?

According to the reviewers/editors’ comments, we checked whether the citations match our statements. When we found the best manuscript that fit our description, we replaced previous citations with new citations.